# The MFN2 Q367H variant reveals a novel pathomechanism connected to mtDNA-mediated inflammation

Mashiat Zaman[1], Govinda Sharma[2,*], Walaa Almutawa[2,*], Tyler GB Soule[3], Rasha Sabouny[2], Matt Joel[3], Armaan Mohan[2], Cole Chute[2], Jeffrey T Joseph[4], Gerald Pfeffer[5], Timothy E Shutt[6]

Pathogenic variants in the mitochondrial protein MFN2 are typically associated with a peripheral neuropathy phenotype, but can also cause a variety of additional pathologies including myopathy. Here, we identified an uncharacterized MFN2 variant, Q367H, in a patient diagnosed with late-onset distal myopathy, but without peripheral neuropathy. Supporting the hypothesis that this variant contributes to the patient's pathology, patient fibroblasts and transdifferentiated myoblasts showed changes consistent with impairment of several MFN2 functions. We also observed mtDNA outside of the mitochondrial network that colocalized with early endosomes, and measured activation of both TLR9 and cGAS-STING inflammation pathways that sense mtDNA. Re-expressing the Q367H variant in MFN2 KO cells also induced mtDNA release, demonstrating this phenotype is a direct result of the variant. As elevated inflammation can cause myopathy, our findings linking the Q367H MFN2 variant with elevated TLR9 and cGAS-STING signalling can explain the patient's myopathy. Thus, we characterize a novel MFN2 variant in a patient with an atypical presentation that separates peripheral neuropathy and myopathy phenotypes, and establish a potential pathomechanism connecting MFN2 dysfunction to mtDNA-mediated inflammation.

## Introduction

Mitochondria are highly dynamic cellular structures that fuse, divide, move about the cell, and interact with other organelles. The integral outer mitochondrial membrane (OMM) proteins Mitofusin 1 (MFN1) and Mitofusin 2 (MFN2) mediate fusion of the OMM. MFN2, though named for its role in mitochondrial fusion, is also actively involved in several other dynamic processes (1): it mediates contacts between mitochondria and the ER (2), referred to here as mito-ER contact sites (MERCs); is involved in mitochondrial interactions with lipid droplets (3), early endosomes (4), and the nucleus (5); and also plays an active role in mitochondrial autophagy (mitophagy) (6, 7).

Impairments in mitochondrial dynamics are linked to genetic neuropathies, with pathogenic variants in MFN2 that are typically associated with the axonal peripheral neuropathy Charcot–Marie–Tooth disease subtype 2A (CMT2A) (8). Characterized by weakness, numbness, and pain in the distal regions of the body, CMT2A is often an early-onset progressively worsening disease (9). In addition to peripheral neuropathy, patients with certain MFN2 variants can present with other pathogenic phenotypes, such as myopathy (10, 11, 12), optic atrophy, sensorineural hearing loss, ataxia (13), or multiple symmetric lipomatosis (14, 15, 16). How and why different variants in MFN2 lead mechanistically to pathology is not fully understood, though it is likely not as simple as reduced mitochondrial fusion (17).

In addition to its critical role in neurons, MFN2 is also important for muscle function. Muscle-specific MFN2 KO results in muscle atrophy, which is attributed to loss of mitochondrial fusion and subsequent instability of the mitochondrial genome (mtDNA) (18). On top of the depletion of mtDNA copy number that is thought to contribute to respiratory deficiency, mtDNA can accumulate point mutations and deletions in the muscle tissue of these KO mice. The role of MFN2 in maintaining mtDNA is further supported by the fact that certain pathogenic MFN2 variants show a reduction in mtDNA copy number (19, 20) or lead to mtDNA deletions (21). However, highlighting our incomplete understanding of the pathology caused by MFN2 deficiency and the role played by mtDNA, not all pathogenic MFN2 variants impair mitochondrial fusion, impact mtDNA, or cause respiratory deficiency (17).

---

[1]Department of Biochemistry & Molecular Biology, Cumming School of Medicine, Hotchkiss Brain Institute, University of Calgary, Calgary, Canada    [2]Department of Biochemistry & Molecular Biology, Cumming School of Medicine, University of Calgary, Calgary, Canada    [3]Department of Neuroscience, Cumming School of Medicine, Hotchkiss Brain Institute, University of Calgary, Calgary, Canada    [4]Hotchkiss Brain Institute, Department of Clinical Neurosciences, Department of Pathology, Cumming School of Medicine, University of Calgary, Calgary, Canada    [5]Hotchkiss Brain Institute, Department of Clinical Neurosciences; and Alberta Child Health Research Institute, Department of Medical Genetics, Cumming School of Medicine, University of Calgary, Calgary, Canada    [6]Departments of Medical Genetics and Biochemistry & Molecular Biology, Cumming School of Medicine, Hotchkiss Brain Institute, Snyder Institute for Chronic Diseases, Alberta Children's Hospital Research Institute; University of Calgary, Calgary, Canada

Correspondence: gerald.pfeffer@ucalgary.ca; timothy.shutt@ucalgary.ca
*Govinda Sharma and Walaa Almutawa contributed equally to this work

---

 

Beyond its role of encoding OXPHOS proteins, mtDNA is now also recognized to play an important role as an inflammatory signalling molecule when released from mitochondria (22, 23). Examples of inflammatory signalling pathways that detect mtDNA-mediated inflammation include TLR9 (4, 19), cGAS-STING (24, 25), and the inflammasome (26, 27). Notably, a recent study showed that inducible muscle-specific loss of MFN1 causes mtDNA release via early endosomes, leading to activation of TLR9/NF-κB inflammation and myopathy (4). Importantly, pharmacological inhibition of this inflammation improved the muscle atrophy and physical performance in these mice, implicating sterile inflammation in the muscle pathology of these mice. Nevertheless, the role of mtDNA-mediated inflammation in the pathology of pathogenic MFN2 variants has not been described to date.

Here, we report a novel MFN2 variant, Q367H, in a patient diagnosed with late-onset distal myopathy. Functional characterization of patient fibroblasts, transdifferentiated myoblasts, and a KO re-expression cell model reveals mtDNA-mediated inflammation likely explains the patient myopathy, thus expanding our understanding of the mechanisms by which MFN2 dysfunction contributes to pathology.

## Results

### Patient report, whole-exome sequencing, and protein expression for MFN2 Q367H variant

A male patient of 73 yr presented with a 2-yr history of slowly progressive weakness affecting his lower extremities. Initial symptoms included difficulty walking uphill, and he eventually developed difficulty with stairs and limitations with walking on the ground level and rising from a chair. There were no cramps, fasciculations, or contractures. There was no sensory loss or coordination. Dysarthria, dysphagia, ptosis, or diplopia was not present.

The patient was medically well and did not have any history of cardiac disease. The mother of the proband developed weakness in her lower legs in her 60s, with progressive limitations in her mobility. Apparently, she was diagnosed with CMT disease, but her clinical profile was not available for review. The mother died at the age of 81 because of congestive heart failure. The mother's two siblings aged 80 and 78, and her other two sons of ages 45 and 43 were all unaffected. Clinical examination of the patient demonstrated a cognitively normal individual. The cranial nerve examination was normal. Motor examination revealed MRC grade 4+ strength for ankle dorsiflexion and 4+ plantar flexion bilaterally. Proximal and axial muscle testing was normal, except for Achilles tendon reflexes. The sensory examination was normal. Creatine kinase enzyme activity was elevated to a maximum of 455 U/l (reference range ≤ 195 U/l). The muscle biopsy showed necrotic and regenerating myofibres, myofibre hypertrophy, excessive endomysial and perimysial fibrosis, and extensive infiltration by adipose tissue. Immunostaining for dystrophin, α/γ-sarcoglycan, spectrum, α-laminin, desmin, and α-β-crystallin was unremarkable (Fig 1A–F). There was no inflammatory cell infiltrate. For the magnetic resonance imaging examination, upper and lower legs

revealed fatty infiltration of the adductor longus, gluteus minimus, tibialis anterior, and inferior gastrocnemius–soleus bilaterally (Fig 1G and H).

The case was interpreted as a chronic myopathy with nonspecific muscle pathology. A genetic cause was suspected given the family history of a similar condition in his mother, and lack of inflammation on muscle biopsy. Clinical testing with a gene panel encompassing 200 genes associated with myopathy, neuropathy, and neuromuscular disorders identified a previously uncharacterized heterozygous variant in the *MFN2* gene, c.1101G>C, p. (Gln367His) (Table 1 and Fig 2A). Research-based exome sequencing again identified the MFN2 variant, as well as heterozygous variants of unknown significance in *SYNE1* and *DYSF* (Table 1). Although pathogenic variants in *SYNE1* can be associated with autosomal dominant neuromuscular disease, this patient variant was not considered to be clinically relevant for the following reasons: (a) the variant is not in the same domain as pathogenic variants previously associated with autosomal dominant disease, (b) *SYNE1* variants in this region of the gene have only been associated with autosomal recessive disorders, whereas our patient is heterozygous, (c) no characteristic nuclear abnormalities associated with pathogenic variants in *SYNE1* (28) were observed in patient fibroblasts under microscopy, and (d) the myopathy phenotype is not in keeping with the *SYNE1*-related disease phenotype, which includes proximal weakness and contractures. Meanwhile, the *DYSF* variant was not considered to be an explanation for this patient's disease because it is a single heterozygous variant in a condition caused by biallelic variants in DYSF, and we are not aware of any reports of dysferlinopathy associated with only a single pathogenic variant. Thus, based on strong suspicion of the relationship between the MFN2 variant and the patient phenotype, we proceeded with the studies described in this report using patient fibroblast cells.

### Characterization of patient fibroblasts

To investigate whether the Q367H variant contributes to the patient phenotype, we began by examining skin fibroblasts obtained from the patient. Immunoblotting for mitochondrial fusion proteins revealed no significant changes in the expression of MFN2, MFN1, or the inner mitochondrial membrane fusion protein OPA1 (Fig 2C–G). Next, to determine whether this MFN2 variant may contribute to the pathology and to gain potential insight into the disease mechanism(s), we characterized patient fibroblasts for alterations to a variety of MFN2-mediated functions.

### Mitochondrial network morphology and respiration in MFN2 Q367H fibroblasts

First, as fibroblast cells from CMT2A patients with MFN2 variants sometimes have fragmented mitochondria (17), we examined mitochondrial morphology as a potential downstream readout of altered mitochondrial fusion (Fig 3A–C). No changes were observed under standard culture conditions. However, glucose-free growth conditions supplemented with galactose, which forces cells to use mitochondria for energy rather than glycolysis, can be used detect to morphology changes in fibroblasts from patients with MFN2 variants (29). Consistent with decreased mitochondrial fusion and

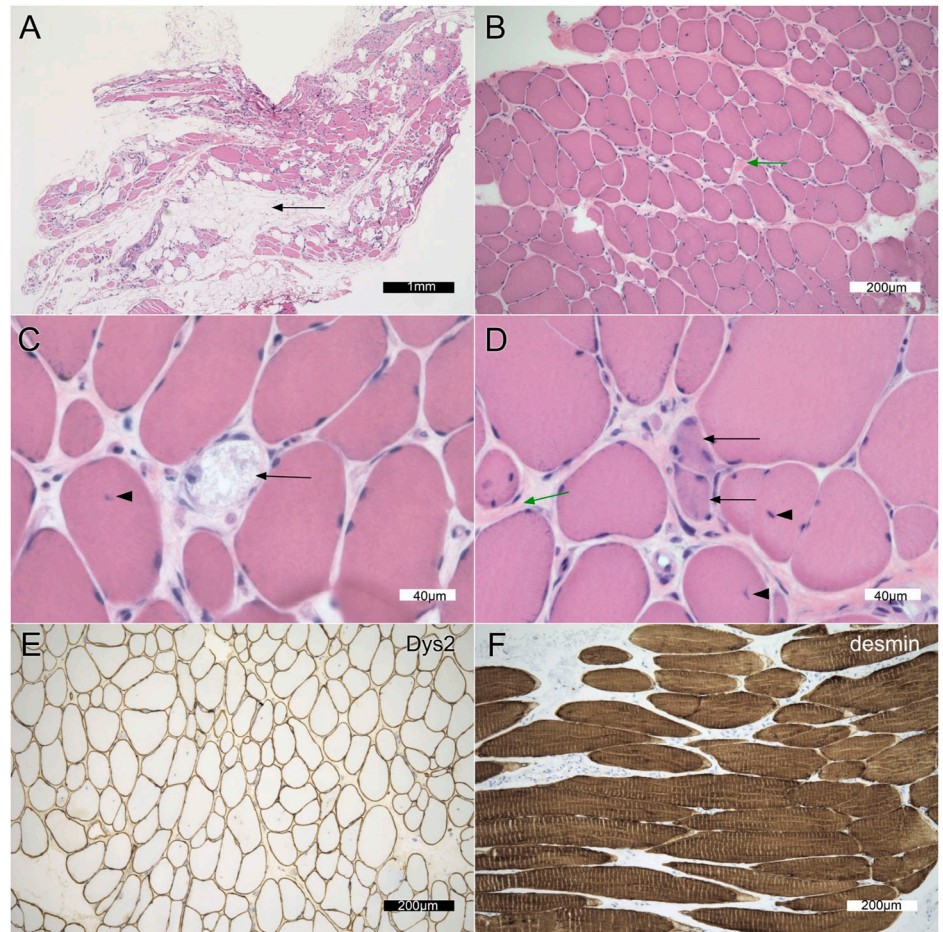

**Figure 1. Muscle pathology findings.**
**(A, B, C, D, E, F)** Representative images of the muscle biopsy (A, B, C, D, E, F) including two formalin-fixed, paraffin-embedded slides (A, F) and four snap-frozen muscle sections (B, C, D, E). Distances are represented in the scale bars. **(A)** (haematoxylin–eosin; H&E) demonstrates extensive adipose tissue ("fatty replacement") in the biopsy (black arrow). This was less obvious in the frozen tissue, which had been dissected away from obvious fat. **(B)** (H&E) illustrates the variation in myofibre size, as well as the increased connective tissue between myofibres (endomysial fibrosis; green arrow). **(C)** Biopsy had several necrotic fibres (panel (C), H&E; black arrow) and increased numbers of internalized nuclei (black arrowhead). In panel (D) (H&E) are two basophilic regenerating fibres (black arrows), internalized nuclei (black arrowheads), and increased endomysial connective tissue (green arrow). **(E)** Dystrophin immunoperoxidase stains the myofibre sarcolemma (Dys2 illustrated in panel (E)). Dys1 and Dys3 are similar (data not shown). **(F)** Desmin immunoperoxidase in longitudinal sections (panel (F)) illustrates the repetitive sarcolemmal units (stripes in each fibre) but does not stain significant sarcoplasmic deposits. **(G)** MRI image at the level of the legs demonstrating predominant atrophy in white arrows: gastrocnemius (i), soleus (ii), and tibialis anterior (iii) bilaterally. **(H)** MRI image of upper thighs demonstrating predominant atrophy of adductor longus and gluteus minimus.

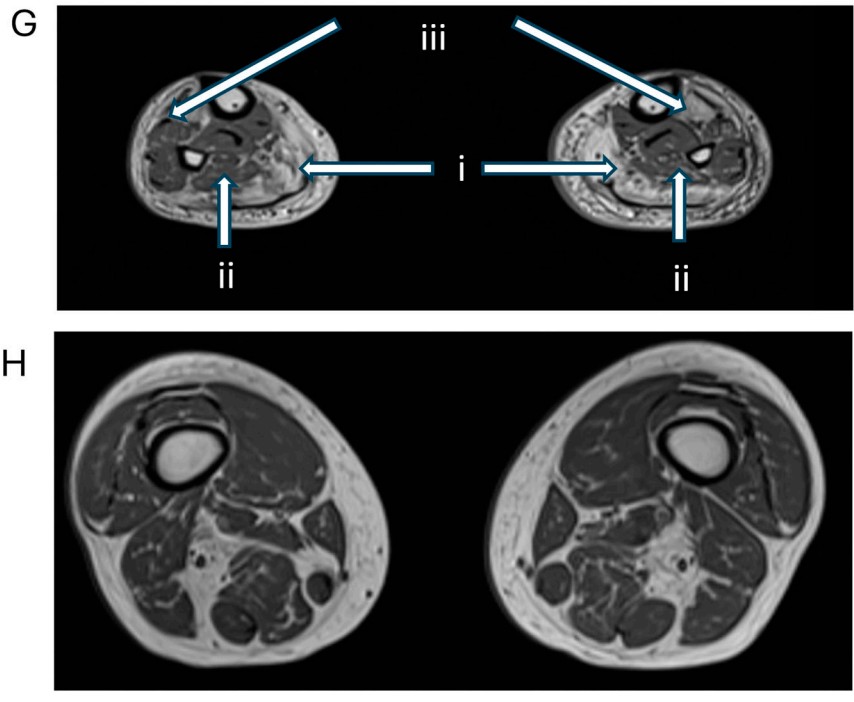

**Table 1.  Genetic variants of note identified in the patient.**

| Gene | Disease/ Inheritance pattern | Coding variant | Amino acid change | rsID | Max reported MAF | ClinVar | Zygosity | PolyPhen2 prediction | MutationTaster prediction |
|---|---|---|---|---|---|---|---|---|---|
| MFN2 | CMT neuropathy 2A; AD | c.1101G>C | p.Gln367His | rs373211062 | 0.00024 | VUS, conflicting interpretations | Heterozygous | B | D |
| DYSF | LGMD 2B; AR | c.5180G>C | p.Gly1727Ala | rs146153532 | 1.16E-04 | VUS | Heterozygous | P | D |
| SYNE1 | EDMD 4; AD/AR | c.9934G>A | p.Asp3312Asn | rs147281213 | 2.33E-04 | VUS | Heterozygous | D | D |

AD = autosomal dominant; AR = autosomal recessive; B = benign; CMT = Charcot–Marie–Tooth; D = deleterious; EDMD = Emery–Dreifuss muscular dystrophy; LGMD = limb–girdle muscular dystrophy; MAF = minor allele frequency; P = possibly tolerated; VUS = variant of uncertain significance.

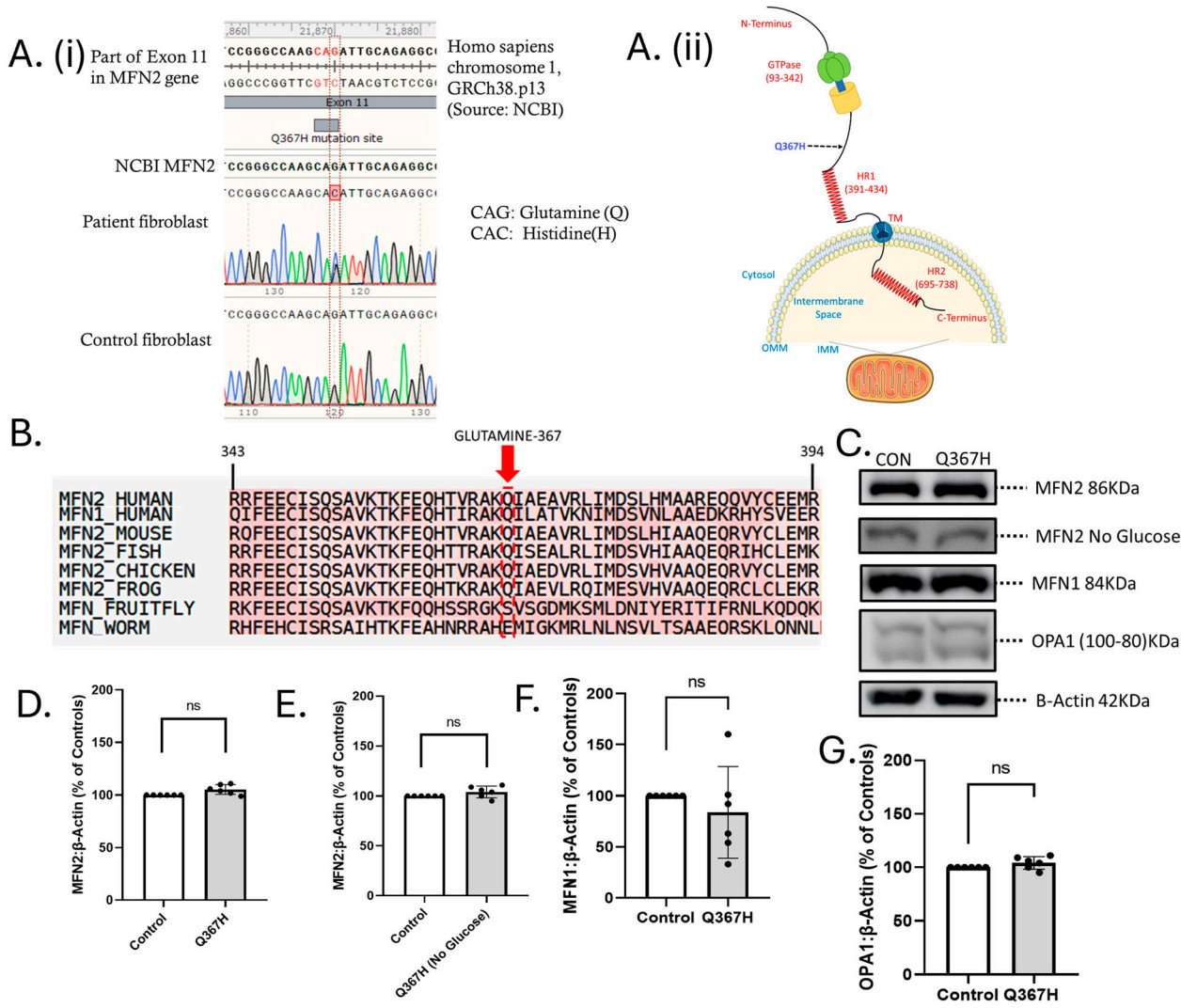

**Figure 2.  Initial characterization of the MFN2 Q367H variant.**
**(A)** (i) Sanger sequencing results showing single nucleotide polymorphism in patient fibroblasts, transition from guanine to cytosine. (ii) Topology diagram of the MFN2 protein with domains and localization indicated. The site of the Q367H variant is shown in the diagram. **(B)** Amino acid sequence alignment using the T-COFFEE tool. **(C)** Western blot analysis comparing the expression of mitochondrial fusion protein in control versus patient fibroblasts, normalized to β-actin. **(D, E, F, G)** Comparison of protein expression levels (D) MFN2 (glucose media), (E) MFN2 (glucose-free/galactose-supplemented media), (F) MFN1, and (G) OPA1 levels in control versus patient fibroblasts. Data are indicative of the mean ± SD. Statistical comparisons were made by an unpaired $t$ test (ns = not significant [$P > 0.05$]).

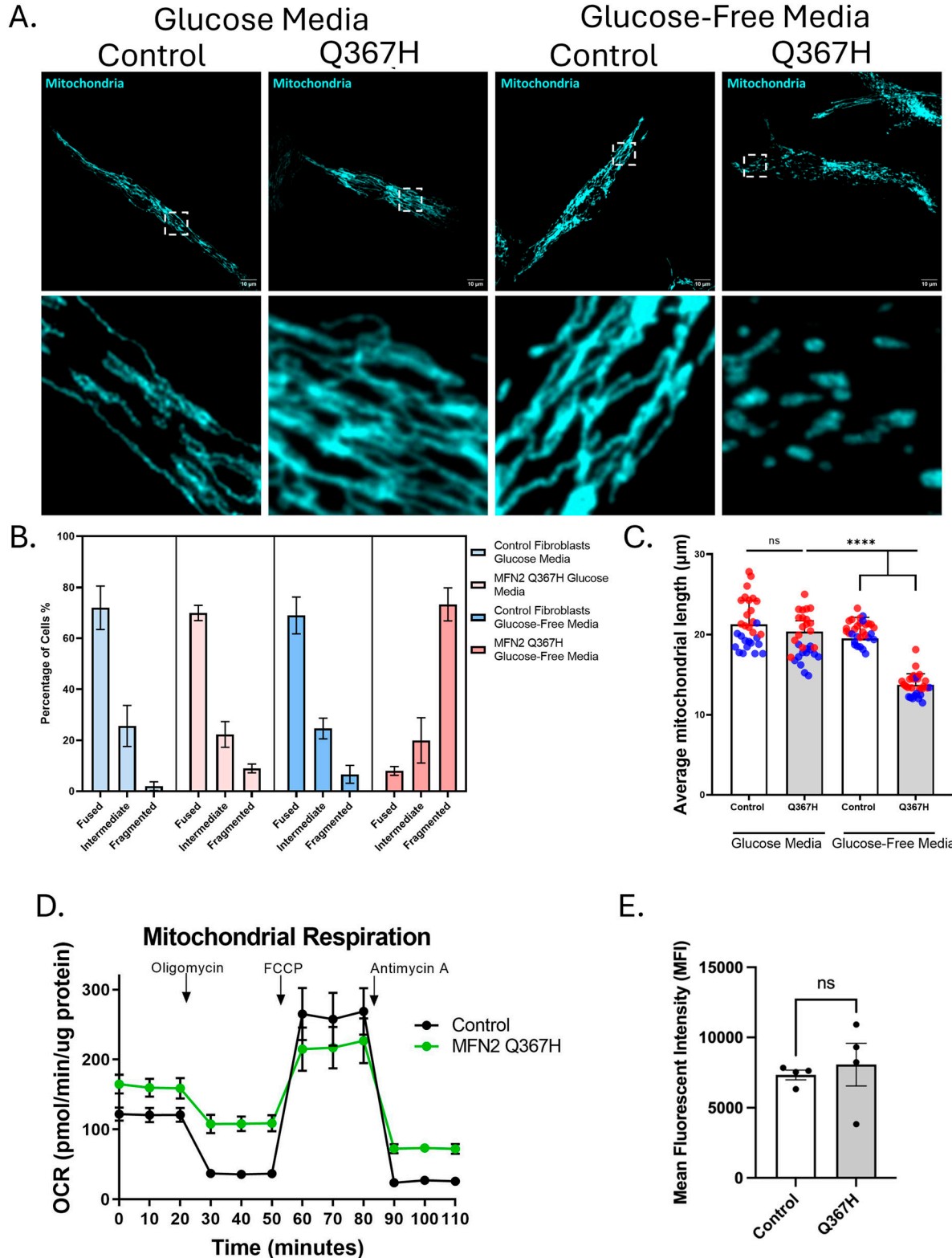

**Figure 3. Changes in mitochondrial morphology and respiratory function.**
**(A)** Representative confocal images (top) with zoomed inset images (bottom) showing mitochondrial network morphology in control versus patient fibroblasts stained with TOMM20. **(B)** Qualitative scoring of proportion of cells showing fragmented/intermediate/fused mitochondrial network. Bars indicate the mean ± SD. **(C)** Quantitative analysis of mean mitochondrial branch length in control versus patient fibroblasts using glucose or glucose-free/galactose-supplemented media; bars show the mean ± SD, and colours indicate biological replicates. **(D)** Oxygen consumption rate (OCR) assay on control and patient fibroblasts using Seahorse XFe24; bars indicate the mean

mitochondrial dysfunction, Q367H patient fibroblasts grown in glucose-free/galactose-supplemented media showed a marked increase in the number of cells with fragmented mitochondrial networks, as well as a decrease in average mitochondrial length. Impairment of mitochondrial function in patient fibroblasts was further reinforced by functional studies of oxygen consumption. Patient fibroblasts had increased basal oxygen consumption but reduced maximal oxygen consumption (Fig 3D). We also measured mitochondrial membrane potential using TMRE, but did not observe any significant differences between control and patient cells (Fig 3E). As mitochondrial fragmentation alone does not indicate MFN2 dysfunction (e.g., reduced mitochondrial health in general can lead to mitochondrial fragmentation), we also explored other functions performed by MFN2 to provide additional evidence for MFN2 dysfunction.

### Perturbated mitochondrial nucleoids in MFN2 Q367H patient fibroblasts

Mitochondria house their own genome, mtDNA, which is present in hundreds of copies throughout the mitochondrial network and is organized into nucleoid structures. As mtDNA nucleoids can be altered by impairments to mitochondrial fusion (30), we sought to examine mtDNA in Q367H patient fibroblasts (Fig 4A and B). Similar to mitochondrial morphology, there were no changes to mtDNA nucleoid size or abundance in standard media. However, under glucose-free/galactose-supplemented conditions, patient fibroblasts had a decrease in the average number of mitochondrial nucleoids per cell, whereas the average nucleoid size was increased (Fig 4A and C). Of note, in both growth conditions, a small number of mtDNA nucleoids were observed outside the mitochondrial network in patient fibroblasts (circled in Fig 4A). Meanwhile, the mtDNA copy number did not change under any conditions (Fig 4D and E), nor were any mtDNA deletions detected (Fig 4F). Observing fewer, larger nucleoids without changes in mtDNA copy number is likely due to nucleoid clustering, a phenotype often observed with impaired mitochondrial fusion (30). As such, the mtDNA studies are consistent with reduced fusion leading to mitochondrial fragmentation in the Q367H cells when grown in glucose-free/galactose-supplemented media.

### MFN2 Q367H fibroblasts show a reduced number of Mito-ER contact sites

MFN2 also mediates MERCs (2). As an estimate of MERCs, we used a proximity ligation assay to detect regions where the mitochondria and ER are within ~40 nm. Colocalization analysis of the PLA signal with either mitochondria or ER (Fig S1) was used to validate the PLA signal as representative of MERCs. No absolute changes in the number or size of the PLA signals were observed in standard media. However, under glucose-free conditions, Q367H cells had fewer PLA puncta, though these puncta were slightly larger (Fig 5A–C). As

changes in MERCs could be due to global changes in mitochondrial abundance, we also measured the area of mitochondria (mitochondrial footprint). Importantly, we do not observe significant differences between mitochondrial footprint in control and patient cells (Fig 5D). However, when the MERC PLA signal is normalized to the mitochondrial footprint, we see a slight reduction in the number of MERCs in the patient fibroblasts in standard conditions, which is further exaggerated under glucose-free conditions (Fig 5E). Together, these data indicate perturbations in MERCs in the Q367H fibroblasts that are consistent with MFN2 dysfunction.

### Dysfunction in mitochondrial fatty acid import leads to lipid droplet accumulation in patient fibroblasts

Given the role of MFN2 in mediating mitochondrial–lipid droplet interactions, and the fact that lipid droplets are altered in fibroblasts from patients with MFN2 variants (17), we next explored lipid droplets in the Q367H fibroblasts. Interestingly, we see a significant increase in the number and size of cellular lipid droplets in patient fibroblasts, grown in both standard and glucose-free/galactose-supplemented media (Fig 6A–D). To explore the mechanism underlying the increased accumulation of lipid droplets, we examined lipid transfer from lipid droplets to mitochondria using an assay that takes advantage of the BODIPY 558/568 fatty acid precursor (31). In this assay, the fluorescently labelled lipid is loaded into lipid droplets during a 24-h incubation, and its transfer to mitochondria can be monitored over time by microscopy. In control cells in standard conditions, some fatty acid precursor transfers into the mitochondrial network (Fig 6E and F), with higher transfer under glucose starvation conditions (without galactose supplementation). Notably, in patient fibroblasts, no fatty acid trafficking from lipid droplets to mitochondria is observed under either growth condition (Fig 6E and F). These findings suggest that the excess lipid droplets seen in the patient fibroblasts are due to an accumulation of lipids because of a lack of fatty acid transfer from lipid droplets to mitochondria.

### MFN2 Q367H patient cells show mtDNA-mediated inflammation

Following up on observation of nucleoids outside of the mitochondrial network (Fig 4), we looked for evidence of mtDNA-mediated inflammation, which is linked to impaired mitochondrial fusion and can cause myopathy (4). First, to look for signs of an inflammatory response, we examined the expression of several inflammation genes via qRT–PCR. Notably, even when cells were grown in standard glucose conditions, we observed an elevation of genes that mediate TLR9 signalling (i.e., ASC/NLRP3/S100A9) and downstream cytokines of the TLR9 pathway (i.e., IL6/TNF-$\alpha$), as well as genes downstream of the cGAS-STING pathway (INF-$\alpha/\beta$) (Fig 7A). Similar inflammatory signalling was observed when fibroblasts were grown in galactose media (Fig S2). To determine whether the increased expression of these genes was due to the TLR9 or cGAS-

± SD (n = 12 replicates). Statistical comparisons were made by an unpaired $t$ test. **(E)** Mitochondrial membrane potential is reported as mean fluorescence intensity (MFI) of TMRE signal in live cells measured by flow cytometry. Bars indicate the mean ± SEM; points indicate biological replicates, unpaired $t$ test (ns = not significant ($P > 0.05$); ****$P \leq 0.0001$).

**Figure 4. Characterization of mtDNA nucleoids, mtDNA copy number, and mtDNA deletions.**
**(A)** Representative confocal images showing mitochondria (TOMM20; grey) and mitochondrial nucleoids (dsDNA antibody; red) in patient and control fibroblasts under glucose and glucose-free/galactose-supplemented nutrient conditions. Circles indicate the presence of mtDNA nucleoids outside the mitochondrial network. **(B)** Number of nucleoids per cell; mean is shown, with colours indicating biological replicates. **(C)** Average size of mitochondrial nucleoids; violin plots show median and interquartile ranges. **(D, E)** qPCR data showing mitochondrial DNA copy number in (D) glucose and (E) glucose-free/galactose-supplemented media; bars indicate mean

STING signalling pathways, fibroblasts in standard media were treated with chloroquine to inhibit TLR9 signalling (32), or RU.521 (33) to inhibit cGAS (Fig 7A). Strikingly, chloroquine reduced TLR9 targets, whereas RU.521 reduced cGAS-STING targets. The fact that each treatment significantly reduced expression of the relevant genes to the levels seen in the control patient fibroblasts confirms that both TLR9-NF-κB and cGAS-STING signalling are elevated in the patient fibroblasts and that these are the primary pathways driving inflammation.

Given the link to TLR9 signalling, which detects mtDNA in endosomes, we also looked to see whether mtDNA outside the mitochondrial network colocalized with Rab5C-labelled early endosomes. Remarkably, we observed colocalization of extra-mtDNA with early endosomes, including enlarged endosomes harbouring several mtDNA nucleoids (Fig 7B and C). Overall, these findings point to a mtDNA-mediated sterile inflammatory response, mediated through localization of mtDNA into early endosomes and the cytosol (Fig 7D).

### MFN2 Q367H patient transdifferentiated myoblasts show elevated inflammation, disrupted MERCs, and reduced mtDNA nucleoid content

To examine the cellular and inflammatory phenotypes more closely in a relevant cell type, we transdifferentiated control and patient fibroblasts into myoblasts for additional characterization. Upon transdifferentiating, we still observed increased inflammatory signalling (Fig 8A). However, compared with fibroblast cells, patient myoblasts exhibited an additional threefold increase in cGAS-STING interferons, and a 10-fold increase in the TLR9 inflammatory signalling, indicating that these pathways are even more strongly activated in patient muscle cells. Finally, we examined some basic parameters of MFN2 functions in our transdifferentiated patient myoblasts. First, we found no changes in gross mitochondrial morphology or the average length of mitochondrial branches in patient myoblasts (Fig 8B and C). Second, patient myoblasts had fewer MERCs, which were also smaller in size (Fig 8D–F). Finally, the mtDNA nucleoid analysis found a reduction in the number of nucleoids per cell in patient myoblasts, as well as a decrease in the size of the nucleoids (Fig 8G–I). Notably, consistent with findings in patient fibroblasts, a subset of these nucleoids could be found outside of the mitochondrial network.

### Re-expression of MFN2 Q367H in MFN2 KO cells recapitulates mtDNA release phenotype

To determine whether the MFN2 Q367H variant is sufficient to induce the mitochondrial alterations that we observed in patient fibroblasts, we re-expressed either WT MFN2 or the Q367H variant in U2OS MFN2 KO cells (Figs 9 and S3). As MFN2 overexpression can induce artefacts (34), we first confirmed that MFN2 was expressed at endogenous levels (Fig 9B). Validating our KO re-expression approach, we found that WT MFN2 re-expression restored all the phenotypes we observed in the U2OS KO cells (Figs 9 and S3). In line with the normal mitochondrial morphology observed in patient fibroblasts grown in glucose and myoblasts, the mitochondrial network in the Q367H re-expression cells was also restored (Fig 9A and C), further suggesting this variant is competent for mitochondrial fusion. With respect to the number and size of mtDNA nucleoids, the Q367H re-expression cells fell between the extremes in the WT and MFN2 KO cells (Fig S3A and B). Next, we focused on the unique mtDNA release phenotype seen in the Q367H patient cells. Despite the ability of the Q367H variant to restore mitochondrial morphology, and consistent with our earlier observations in patient cells harbouring the Q367H variant, there was a prominent presence of mtDNA outside the mitochondrial network in the U2OS MFN2 KO cells re-expressing Q367H (Figs 9A and S3C), and this mtDNA was predominantly colocalized with endosomes (Figs 9D and S3D and E). Intriguingly, there were significant differences between MFN2 KO and the Q367H re-expression lines with respect to mtDNA release. For example, although there was slightly more extra-mtDNA in U2OS KO cells, only about half of the mtDNA colocalized with endosomes in the MFN2 KO cells (Fig S3C–E). Moreover, compared with MFN2 KO cells, endosomes in the Q367H re-expression cells were more abundant, larger, and on average closer to mitochondria (Fig S3F–H). Overall, these findings show that although the Q367H variant is competent for mitochondrial fusion, it is sufficient to cause mtDNA release and that mtDNA/endosome colocalization and endosome phenotypes are exacerbated with the Q367H variant compared with the MFN2 KO cells.

## Discussion

We report an individual with a late-onset myopathy phenotype and a previously uncharacterized variant in the *MFN2* gene. We pursued the investigations described in this report based on strong clinical suspicion for this as a potential pathogenic variant. This sequence change in the MFN2 protein replaces glutamine with histidine at codon 367 of the MFN2 protein (Q367H). The glutamine residue is highly conserved, and there is a small physicochemical difference between glutamine and histidine, implying a hypermorphic change. The MFN2 Q367 amino acid is conserved across vertebrate MFN2 sequences, as well as with the MFN1 paralogue (Fig 2B). The MFN2 Q367H variant is present in population databases with a very low allele frequency (highest allele frequency 0.000242 in gnomAD according to dbSNP report rs373211062, accessed 4 June 2024), including one homozygous individual. The variant has an entry in ClinVar (439898, accessed 4 June 2024) and is classified as having conflicting interpretations of pathogenicity, with four entries from clinical testing of patients with neurological phenotypes. In a large cohort study, a patient with early-onset peripheral neuropathy was reported who was compound heterozygous for Q367H and P184R MFN2 variants (35). However, the Q367H MFN2 variant has not been functionally characterized, or previously reported in the literature

± SD. **(F)** Agarose gel electrophoresis showing long-range PCR products used to detect full-length or deleted mtDNA and DNA ladder showing corresponding sizes. Statistical comparisons were made by an unpaired t test (ns = not significant ($P > 0.05$); *$P ≤ 0.05$; ***$P ≤ 0.001$; ****$P ≤ 0.0001$).

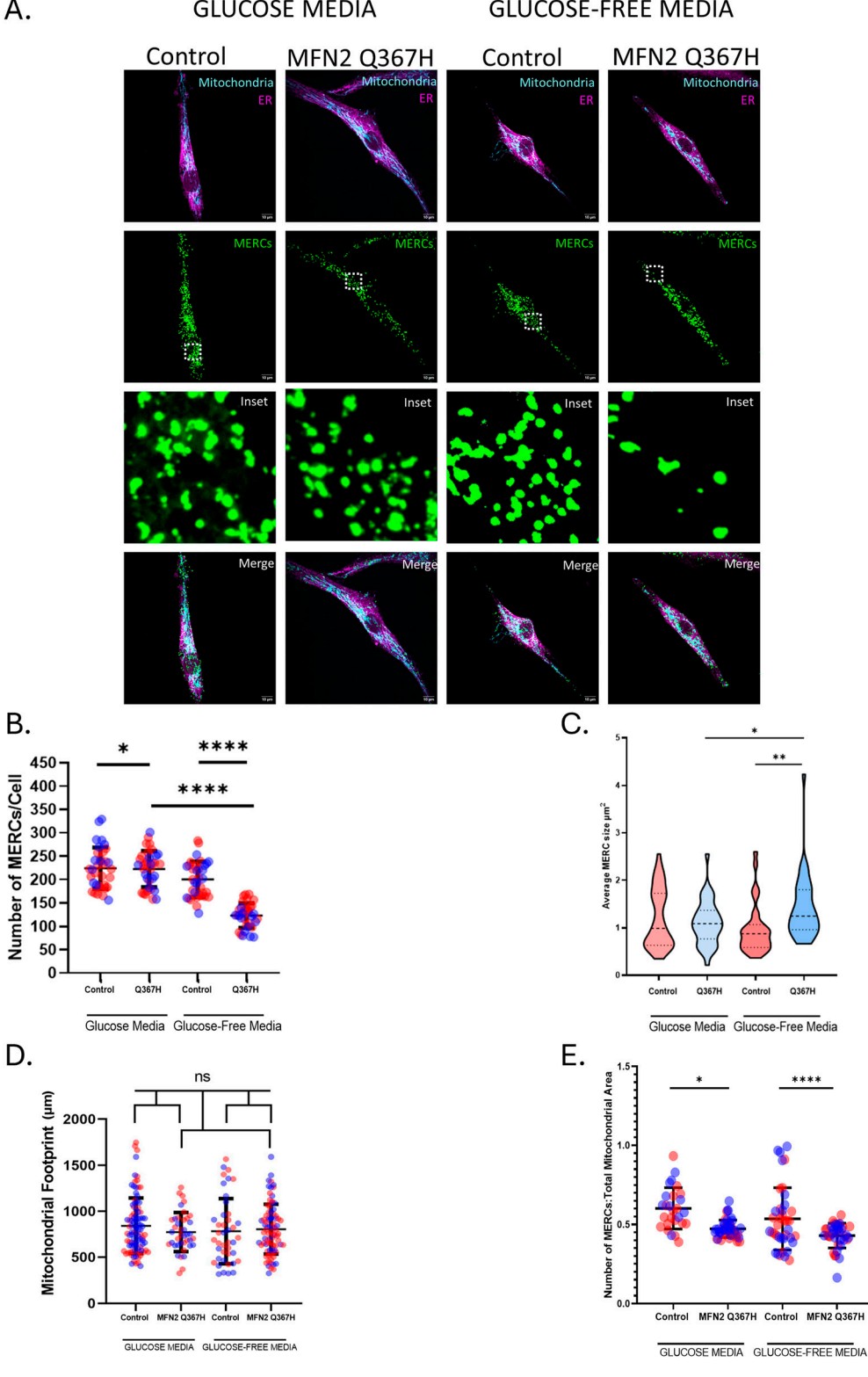

**Figure 5. Mito-ER contact site alterations.**
**(A)** Representative confocal images of mito-ER contact sites (MERCs) using proximity ligation assay (PLA) showing mitochondria (TOMM20; cyan), PLA probes (green), and ER (calnexin; magenta) in glucose and glucose-free/galactose-supplemented media. **(B)** Quantitative analysis of the number of MERCs per cell; lines indicate the mean ± SD, and colours indicate biological replicates. **(C)** Quantitative analysis of the average size of PLA probes; violin plots indicate the median and interquartile range. **(D)** Quantitative analysis of mitochondrial area in each cell type in $\mu m^2$; lines show the mean ± SD, and colours indicate biological replicates. **(E)** Normalized number of MERCs to total mitochondrial area per cell; lines indicate the mean ± SD, and colours indicate biological replicates. $P$-values were determined by an unpaired $t$ test (ns = not significant [$P > 0.05$]; *$P \leq 0.05$; **$P \leq 0.01$; ****$P \leq 0.0001$).

as a standalone disease-causing variant. The family history of the patient's mother suggests autosomal dominant inheritance, which is in keeping with this variant, although confirmation of this variant was not possible in the mother. Furthermore, a different amino acid substitution at this position (Q367P) was associated with a neuropathy presentation (36), providing additional support that

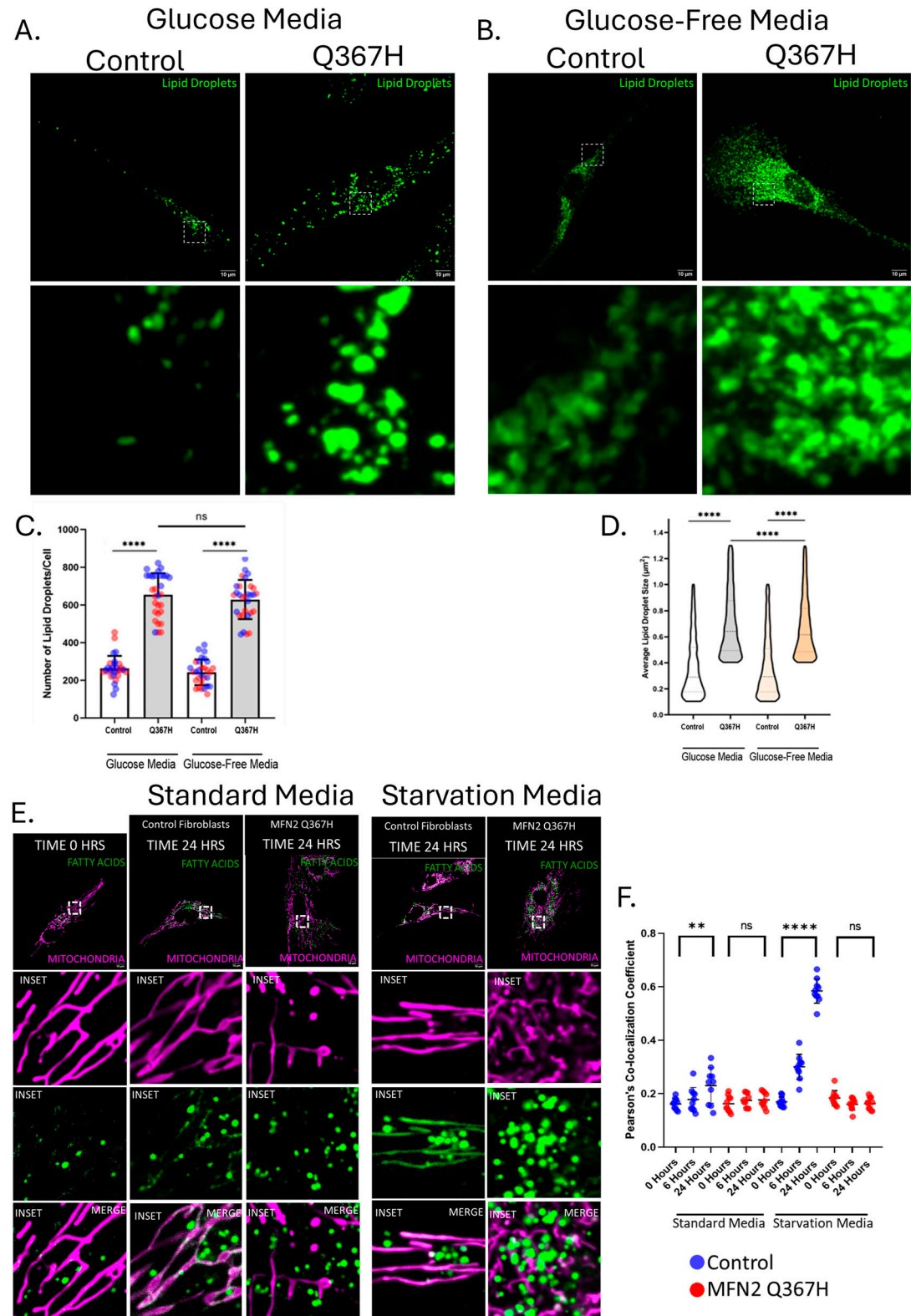

**Figure 6. Changes to cellular lipid droplets.**
**(A, B)** Representative confocal images showing lipid droplets stained with HCS LipidTox Green in (A) glucose and (B) glucose-free/galactose-supplemented media. **(C)** Quantitative analysis of the average number of lipid droplets per cell; bars indicate the mean ± SD, and colours indicate biological replicates. **(D)** Quantitative analysis of the average size of lipid droplets; violin plots show the median and interquartile range. **(E)** Representative live-cell confocal images showing mitochondria (MitoTracker Green) and fatty acids (BODIPY 558/568) at 0 and 24 h in standard or glucose-starved media. **(F)** Quantitative analysis of colocalization between fatty acid and

variants at this position are likely to be associated with this disease. All considered, the Q367H variant is highly plausible as a reduced penetrance pathogenic variant given its extremely low frequency in population databases, its prevalence in individuals with neuro-logical phenotypes, and its presence in published cases from literature. The data from studies in this report point to the deleterious nature of the Q367H variant, and importantly demonstrate the damaging effect of this variant in cells transdifferentiated to the muscle lineage.

This case illustrates many of the challenges when conducting research with people affected by genetic conditions with very late onset. Information from control population databases may be misleading if individuals are presymptomatic for a very-late-onset disease, or when phenotypes are very mild and slowly progressive. Even for paediatric-onset disease, pathogenic variants appear in a substantial proportion of control individuals (2.8% in one study) (37), and this number may be comparable or even higher for very-late-onset disorders, such as the one described in this report. Future research will improve our understanding of genetic variants and disorders with very late onset and variable penetrance, but identification and investigation of cases is difficult because they may be mistaken for degenerative or acquired conditions (e.g., prior work showing diagnoses of genetic spastic paraplegia in patients initially diagnosed with sporadic primary lateral sclerosis) (38). Other factors include the lack of availability of family members for segregation of variants (i.e., parents have expired by the time the patient is diagnosed, or family members are estranged, or the disease status is not yet known for younger family members). In this respect, we acknowledge the limitations of this study, which only includes a single affected participant. However, we hope to bring attention to the fact that the investigation of genetic disorders is important to our broader understanding of genetic diseases and operates with the above-mentioned special challenges. Furthermore, our findings from the re-expression of the variant in a KO cell line show the importance of functional characterization to confirm potential pathogenic variants.

Despite the fact that over 150 MFN2 variants have been reported in patients with CMT2A, we still do not have a complete understanding of how MFN2 dysfunction contributes to disease. The situation is complex, as MFN2 has multiple functions in the cell, and CMT2A can have additional pathological phenotypes beyond peripheral neuropathy. Moreover, only a handful of MFN2 variants have been characterized functionally, and, even then, often only for one or two of MFN2's known functions (17). Thus, a better understanding of how MFN2 variants affect MFN2 functions is essential to understanding how MFN2 contributes to disease. Here, our characterization of cells from a patient exhibiting distal myopathy who carried the candidate pathogenic MFN2 Q367H variant provides novel mechanistic insight into MFN2-mediated pathology. The fact that several MFN2-mediated functions are impaired in the patient cells, especially when they are forced to use mitochondria for energy production, supports the notion that the Q367H variant impairs MFN2 function and likely contributes to the patient's

pathology. Moreover, to our knowledge, the association of mtDNA release and inflammation with an MFN2 variant has not been reported previously, and supports a mechanistic pathway by which MFN2 dysfunction contributes to inflammation. Finally, the elevated levels of inflammation in patient-derived transdifferentiated myoblasts further support a novel disease pathomechanism for MFN2-mediated inflammation driving myopathy.

Several lines of evidence support the pathogenicity of the Q367H MFN2 variant. Patient fibroblasts show perturbation to multiple functions performed by MFN2, including mitochondrial fusion, mtDNA nucleoid distribution, MERCs, and cellular lipid droplets. Similarly, characterization of our Q367H re-expression line also provides direct evidence that this variant is not fully functional. With respect to mitochondrial fusion, similar to our findings, several MFN2 variants do not show mitochondrial fragmentation when grown in standard glucose conditions (17). However, culturing fibroblasts in galactose can reveal fragmentation of the mitochondrial network, as reported for MFN2 D210V (10), and as we see for the Q367H variant. This network fragmentation could be caused by reduced fusion and/or increased fission, which can occur in response to a variety of stimuli (39). Given the well-defined role of MFN2 in mitochondrial fusion, the fragmentation is consistent with reduced fusion. Meanwhile, as reduced mitochondrial fusion can also impact mtDNA distribution (30), the changes we see in mitochondrial nucleoids are also consistent with reduced fusion in patient fibroblasts when grown in galactose. Nevertheless, the Q367H re-expression line shows that this variant can restore mitochondrial morphology in MFN2 KO cells. Thus, any impairment in mitochondrial fusion is likely to be minor, and only noticeable under certain conditions, such as when grown in galactose.

Beyond mitochondrial fusion, perhaps the best-characterized function of MFN2 is in regulating MERCs (2). Notably, MERCs in fibroblasts from patients with certain MFN2 variants can show a variety of changes (17), and the reduction in the number and size of MERCs that we observed in patient cells is consistent with MFN2 dysfunction. Intriguingly, altered MERCs could contribute to the patient myopathy, as pathogenic variants in *SEPN1*, another gene implicated in MERC biology, also cause a myopathy phenotype (40).

Also consistent with MFN2 dysfunction and in line with our findings, an increased number and size of lipid droplets in fibroblasts are reported in fibroblast patients with other MFN2 variants (41), as well as upon MFN2 depletion (3). Intriguingly, the lipid droplet phenotype is evident in standard growth conditions when mitochondrial morphology is unaltered. Notably, the mechanism by which impaired MFN2 function leads to increased size and abundance of lipid droplets in patient fibroblasts has not been investigated previously. We envision several possible explanations: decreased transfer of lipids into mitochondria from lipid droplets, or increased transfer of lipids from mitochondria to lipid droplets. Our finding that patient fibroblasts failed to import fatty acids into mitochondria, even under glucose starvation conditions, is in line with the first possibility, and likely reflects the role of MFN2 in

mitochondrial network; colours indicate the cell type (red—control; and blue—MFN2 Q367H), and bars indicate the mean ± SD. *P*-values were determined by an unpaired *t* test (ns = not significant [*P* > 0.05]; **P ≤ 0.01; ****P ≤ 0.0001).

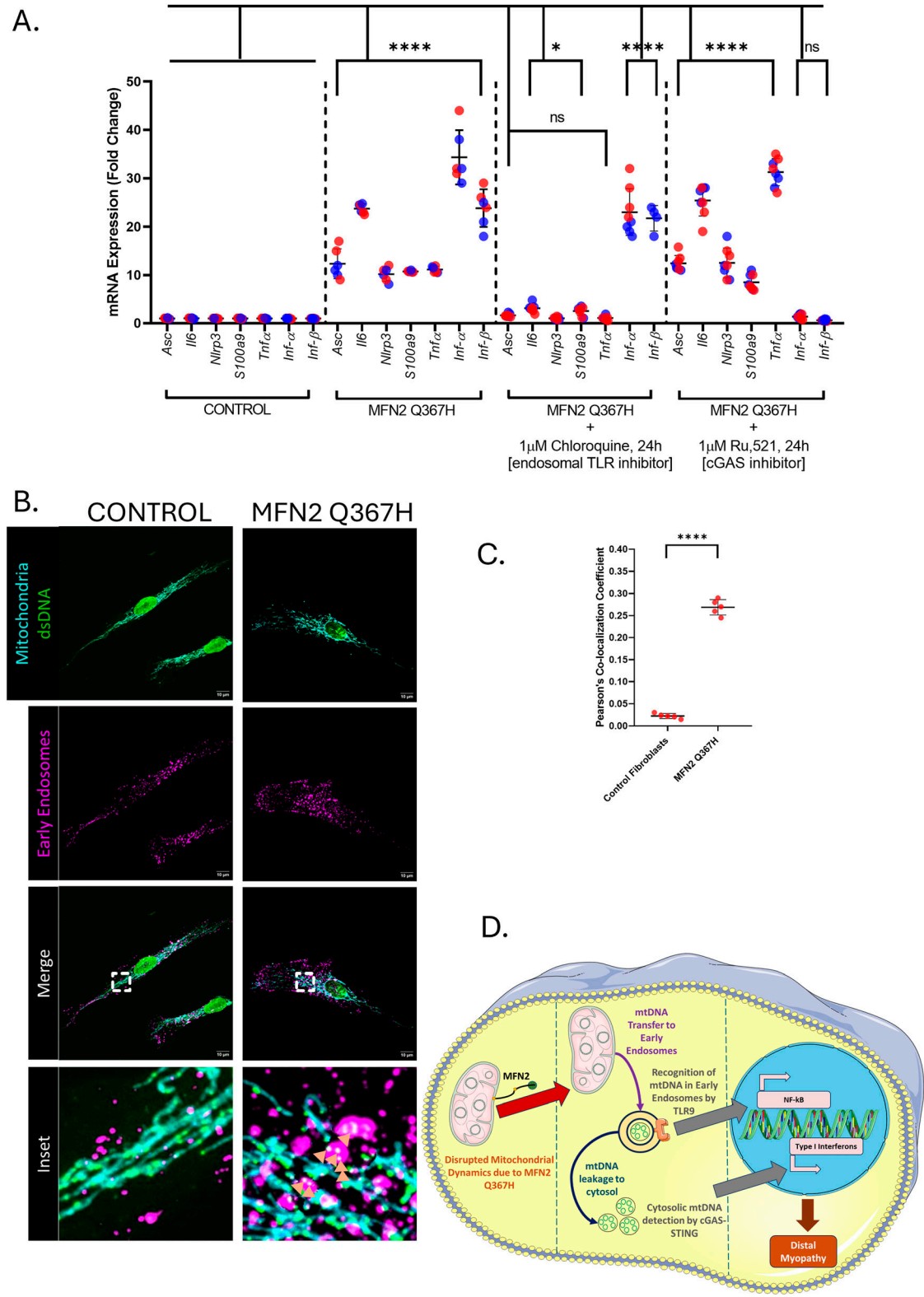

**Figure 7. MFN2 Q367H cells show activation of TLR9 and cGAS-STING inflammatory pathways, and colocalization of mtDNA with early endosomes.**
**(A)** qRT-PCR data showing gene expression for TLR9/NF-κB targets (Asc, IL6, NLRP3, S100a9, TNF) and cGAS-STING targets (Ifn-α, Ifn-β) in control versus patient fibroblasts under glucose conditions. The graph also shows the effects of inhibitors for the TLR pathway (chloroquine) or cGAS-STING pathway (RU.521). Lines indicate the mean ± SD. **(B)** Representative confocal images showing mitochondria (TOMM20; cyan), mitochondrial nucleoids (anti-dsDNA antibody; green), and early endosomes (Rab5C; magenta). Images in the inset bottom right panel highlight the location of mitochondrial nucleoids outside the mitochondria and within the early endosomes,

mediating mitochondrial–lipid droplet interactions (3). Though we expect reduced lipid transfer will also explain the increased lipid droplet accumulation observed in fibroblasts from patients with other MFN2 variants, this remains to be determined.

Characterization of the transdifferentiated patient myoblasts provides further evidence of MFN2 dysfunction and important insight into the muscle myopathy phenotype in the patient. Consistent with MFN2 dysfunction, we see a significant decrease in the number and size of mitochondrial nucleoids, as well as fewer and smaller MERCs in transdifferentiated patient myoblasts. However, we did not observe any changes in mitochondrial network morphology. Notably, the transdifferentiated myoblasts were grown in standard myoblast media, as opposed to the glucose-free/galactose-supplemented media required to elicit differences in the patient fibroblasts. Other differences in growth conditions between transdifferentiated patient myoblasts (e.g., collagen matrix), or the inherent differences between fibroblast and myoblast cell types may also explain differences between patient fibroblasts and transdifferentiated myoblasts. Regardless, the reduced number of nucleoids in the patient cells is consistent with MFN2 dysfunction, as muscle-specific loss of MFN2 leads to mtDNA depletion (18).

The finding of mtDNA release and inflammation in both Q367H patient fibroblasts and transdifferentiated myoblasts is novel in the context of pathogenic MFN2 variants and is likely relevant to the disease mechanism. The mtDNA release and endosome colocalization also observed in the Q367H re-expression cells further show that this variant causes mtDNA release, rather than an undetected difference in the genetic background of the patient. With respect to the inflammatory signals, the fact that inhibitors of both TLR9/cGAS-STING pathways lead to a decrease in their downstream cytokines indicates that both pathways are activated. These findings are also consistent with previous work on impaired mitochondrial fusion and increased inflammation. With respect to MFNs, a recent article by Irazoki et al found that knockdown of either MFN1 or MFN2 in myoblasts could promote TLR9 signalling (4). Arguing that activation of TLR9-mediated inflammation is a general response to loss of mitochondrial fusion, loss of the inner mitochondrial membrane fusion protein OPA1 in muscle leads to TLR9-mediated NF-κB activation (but not cGAS-STING), where it also drives an inflammatory myopathy (19). Thus, it seems that loss of fusion is sufficient to induce mtDNA-mediated TLR9 inflammation in muscle cells that can drive myopathy, a mechanism that may explain the phenotype of the patient harbouring the Q367H variant. Meanwhile, the increased cGAS-STING signalling that we also see in Q367H patient cells is consistent with increased cGAS-STING activity also reported in MFN2-deficient myoblasts (4), as well as in microglia where MFN2 is down-regulated (42). Whether cGAS-STING signalling further perturbs pathology remains to be determined,

though elevated cGAS-STING inflammation is also linked to myopathy (43).

Another interesting question involves how MFN2 loss of function drives mtDNA release to stimulate both TLR9 and cGAS-STING inflammation. In this regard, several recent studies are relevant to endosomal-mediated mtDNA release (44, 45, 46). A model seems to be emerging where, in the context of stalled mtDNA replication intermediates, enlarged nucleoid clusters are transferred directly into early endosomes, where the mtDNA can be detected by TLR9. When this quality control pathway is overwhelmed, it can lead to release of mtDNA from endosomes into the cytosol, and cGAS-STING activation (46). Notably, one mechanism by which loss of mitochondrial fusion is proposed to impact the mtDNA is through imbalanced distribution of mtDNA maintenance proteins (47), which may lead to increased abundance of stalled mtDNA replication intermediates. Thus, mtDNA replication fork stalling in response to impaired mitochondrial fusion may trigger initial mtDNA release into endosomes, and the subsequent TLR9 activation associated with MFN2 dysfunction. The fact that we see extra-mtDNA and inflammatory signalling in patient cells with normal mitochondrial morphology (e.g., fibroblasts grown in standard media and differentiated myoblasts) suggests that even subtle alterations in mitochondrial fusion could be sufficient to impact the mtDNA. Alternatively (or in conjunction), the differences in MERCs, which can demarcate sites of mtDNA replication (48, 49), could also impair mtDNA replication and trigger mtDNA release into endosomes.

Meanwhile, as elevated cGAS-STING is not found with MFN1 or OPA1 depletion (4, 19, 50), the elevated cGAS-STING signalling in myoblasts devoid of MFN2 (4), and seen here in MFN2 Q367H patient cells, suggests another role of MFN2 in mediating mtDNA release beyond just reduced mitochondrial fusion. In this regard, MFN2 can interact with Rab5C on early endosomes (4). As such, it is tempting to speculate that MFN2 has a role in mediating the mitochondrial–endosome interactions required for endosome-mediated removal of stalled mtDNA replication intermediates and that altering this interaction promotes mtDNA escape into the cytosol to activate cGAS-STING inflammation. In this regard, the differences between MFN2 KO and Q367H re-expression cells with respect to endosomal mtDNA and endosome alterations support the notion that the Q367H variant is not just "loss of function" with respect to mtDNA release. Future studies examining the mechanisms of mtDNA release will likely provide more insight into the role played by MFN2 and may benefit from examining the Q367H variant as a means to perturb MFN2 function.

It is worth noting that MFN2 could also influence inflammation through its other functions. For example, there are connections between disrupted MERCs and activation of sterile inflammation, particularly in high energy–demanding tissues such as the heart (51). Meanwhile, relevant to the increased lipid droplet abundance

denoted by the orange arrowheads. **(C)** Quantitative analysis of signal colocalization between mitochondrial nucleoids and early endosomes in patient and control fibroblasts determined by Pearson's coefficient; lines indicate the mean ± SD, and points indicate replicates. **(D)** Pathway model representing mitochondrial dysfunction leading to sterile inflammation. The diagram shows how the MFN2 Q367H variant can alter mitochondrial function, leading to the transfer of mtDNA from the mitochondria to the early endosomes (middle). This can lead to detection by TLR9 and subsequent activation of the NF-κB–mediated inflammation (right). In addition, mtDNA in the early endosomes could leak into the cytosol, leading to detection through the cGAS-STING pathway (middle-bottom), promoting type I interferon–mediated inflammation (right). Collectively, inflammation contributes to the distal myopathy disease outcome in the patient. $P$-values were determined by an unpaired $t$ test (ns = not significant [$P > 0.05$]; *$P \leq 0.05$; ****$P \leq 0.0001$).

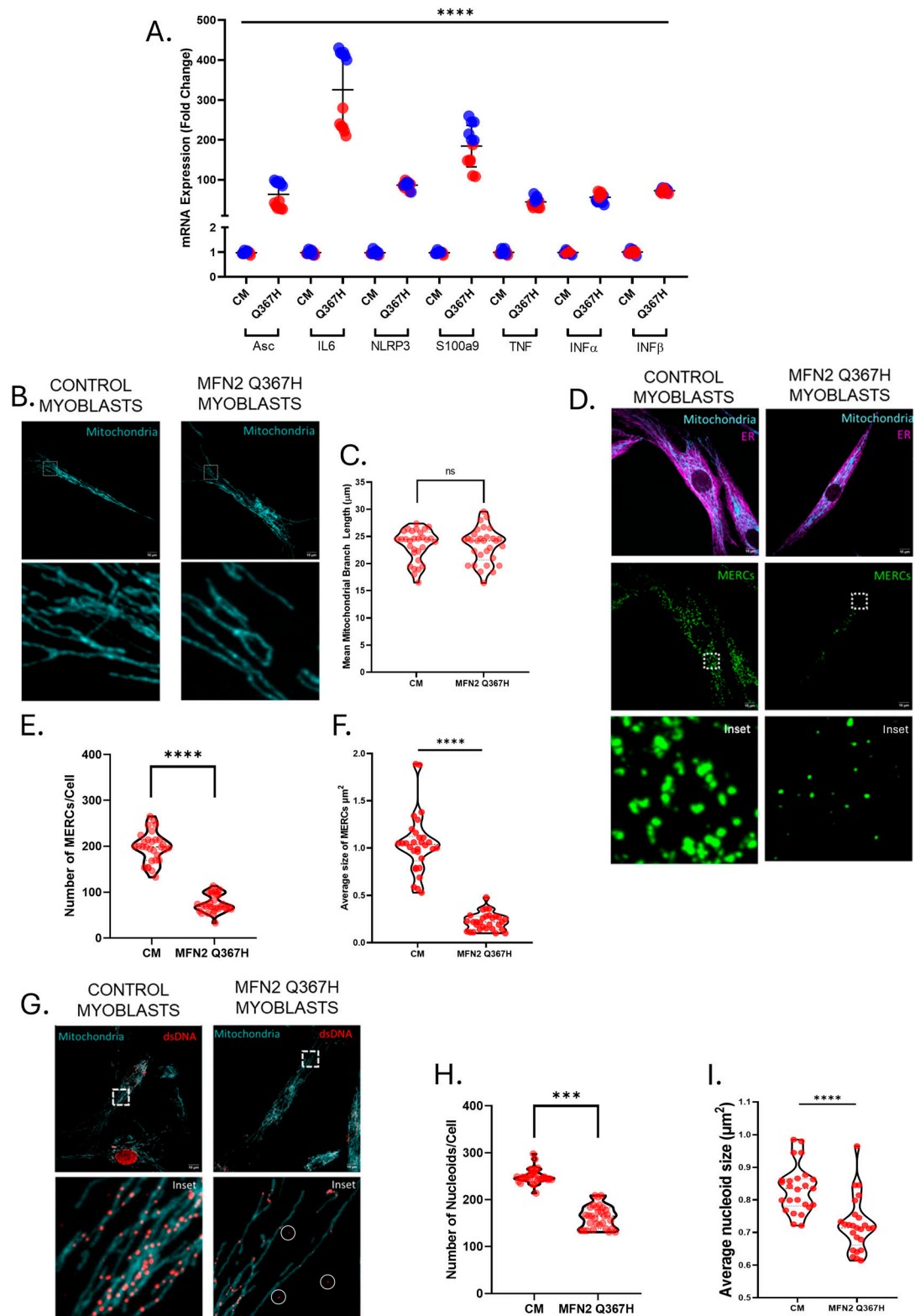

**Figure 8. Altered MFN2 functions in transdifferentiated patient myoblasts.**
**(A)** qRT-PCR data showing gene expression for TLR9/NF-κB targets (Asc, IL6, NLRP3, S100a9, TNF) and cGAS-STING targets (Ifn-α, Ifn-β) in control versus patient myoblasts; lines indicate the mean ± SD.**(B)** Representative confocal images showing mitochondrial network morphology (TOMM20; cyan) in control and patient myoblasts. **(C)** Quantitative analysis of mean mitochondrial branch length in control and patient myoblasts; violin plots show the median and interquartile ranges. **(D)** Representative confocal images of control and patient myoblasts showing PLA probes (representing MERCs; green), mitochondria (TOMM20; cyan), and ER (calnexin;

associated with MFN2 dysfunction and seen even in standard growth conditions with normal mitochondrial morphology, it is notable that lipid droplet biology intersects with inflammation. Specifically, elevated lipid droplet abundance correlates with increased inflammation (52, 53, 54, 55). Overall, our findings are consistent with MFN2 dysfunction contributing to inflammation, which is a possible explanation for myopathy initially described in the proband.

The patient presentation, with late-onset myopathy, but without neuropathy, offers a unique opportunity to separate the pathological outcomes of MFN2 dysfunction by examining the specific mechanism leading to myopathy. Although myopathy has been reported previously in patients with CMT2A (11, 12), it has always been in the context of peripheral neuropathy where it is generally considered to be a downstream consequence of nerve loss. For example, the MFN2 D210V variant has myopathy phenotypes in a three-generational case study, which has accompanying axonal neuropathy (10). On the other hand, arguing against the notion of muscle loss downstream of neuropathy, mitochondrial myopathy is seen in a skeletal muscle–specific MFN2 loss-of-function study in animal models, highlighting the importance of the protein in tissues such as skeletal muscle (18, 56). Meanwhile, in cardiac muscle, the MFN2 R400Q variant is implicated in cardiomyopathy, though no link to inflammation was investigated (57). In addition, the loss of MFN2 protein expression in skeletal muscles correlates with age and could contribute to sarcopenia through a similar inflammatory mechanism (58). Finally, similar to the patient we describe here, a novel ENU-induced Mfn2 mouse model characterized L643P as a variant that causes muscle myopathy without peripheral neuropathy (59). Together, these findings show that MFN2 dysfunction can cause muscle myopathy in the absence of nerve degeneration, as observed in the proband. Future cases of myopathy associated with this variant in MFN2 may benefit from the inclusion of electron microscopic examination of muscle biopsy tissues. EM was not obtained for our case, but may have shown mitochondrial ultrastructural changes to provide further insight into the morphological impacts of the MFN2 variant.

With respect to how MFN2 dysfunction might cause myopathy, in the first report of an MFN2-specific mouse muscle KO, the authors argued that mtDNA depletion was likely a key contributory factor because of reduced oxidative phosphorylation (18). However, more recent work linking MFN1 and OPA1 muscle deficiency to inflammation-mediated myopathy showed that blocking the inflammation was protective in these models (4, 19), arguing for a key role of inflammation in the muscle pathology caused by impaired mitochondrial fusion. Mechanistically, inflammation in the muscle cell niche can lead to myopathy and interfere with muscle stem cell populations (60), as well as affect the expression of myogenic genes (61). Meanwhile, NF-$\kappa$B, a proinflammatory cytokine that is downstream of TLR9 signalling, can cause muscle wasting in mice (62). Building on these findings, it is tempting to speculate that mtDNA-mediated inflammation may also be relevant to idiopathic inflammatory myopathies, a heterogeneous group of autoimmune disorders with varying clinical manifestations, primarily defined by muscle weakness (63, 64). In this regard, we speculate that the mtDNA-mediated inflammation discovered in Q367H patient-derived cells is contributing to the patient myopathy.

Although MFN1 and OPA1 deficiency can cause inflammatory myopathy in mouse muscle KO models, the role of mtDNA-mediated inflammation in MFN2 pathology has not been fully explored. While a mouse knock-in model of the MFN2 K357T variant is linked to augmented neuroinflammation in response to LPS treatment, the role of mtDNA was not investigated (65). Our findings linking the MFN2 Q367H variant to mtDNA release and inflammation thus provide a novel disease mechanism for MFN2 dysfunction. Moreover, although our findings are relevant to the myopathy phenotype in muscle, it is also possible that mtDNA-mediated inflammation contributes to peripheral neuropathy in neurons. Intriguingly, peripheral neuropathy is a common phenotype of several DNA damaging agents (66, 67, 68, 69, 70), such as doxorubicin, which can damage mtDNA and lead to mtDNA-mediated cGAS-STING inflammation (71).

In summary, we observed elevated mtDNA-mediated TLR9/NF-$\kappa$B and cGAS-STING inflammatory signalling in a patient harbouring the uncharacterized MFN2 Q367H variant who displayed late-onset myopathy as a standalone phenotype. Moreover, functional analyses of patient fibroblasts provided several lines of evidence consistent with MFN2 dysfunction. Critically, the characterization of transdifferentiated myoblasts and the elucidation of a novel link to mtDNA-mediated inflammation may mechanistically explain the myopathy phenotype in the patient.

# Materials and Methods

### Ethics statement

The patient provided written informed consent for participation in research and consented to the publication of this report. He was provided with a copy of the draft article before submission. Genetic sequencing and human tissue studies with this participant were part of research protocols (REB15-2763 and REB17-0850) approved by the Conjoint Health Research Ethics Board at the University of Calgary.

### Exome sequencing, variant confirmation, and sequence conservation study

DNA was extracted from blood collected into EDTA tubes using standard protocols. Library preparation proceeded using Ion AmpliSeq Exome RDY Panel (A38264; Thermo Fisher Scientific)

magenta). **(E)** Quantitative analysis of the number of MERCs per cell in control and patient myoblasts; violin plots show the median and interquartile ranges. **(F)** Quantitative analysis of the average size of MERCs per cell in control and patient myoblasts; violin plots show the median and interquartile ranges. **(G)** Representative confocal images showing mitochondria (TOMM20) and mitochondrial nucleoids (dsDNA antibody). Circles indicate mtDNA staining outside the mitochondrial network. **(H)** Quantification of the average number of nucleoids per cell in control and patient myoblasts; violin plots show the median and interquartile ranges. **(I)** Quantification of the average size of mtDNA nucleoids in control and patient myoblasts. Violin plots show the median and interquartile ranges. $P$-values were determined by an unpaired $t$ test (ns = not significant [$P > 0.05$]; ***$P \leq 0.001$; ****$P \leq 0.0001$).

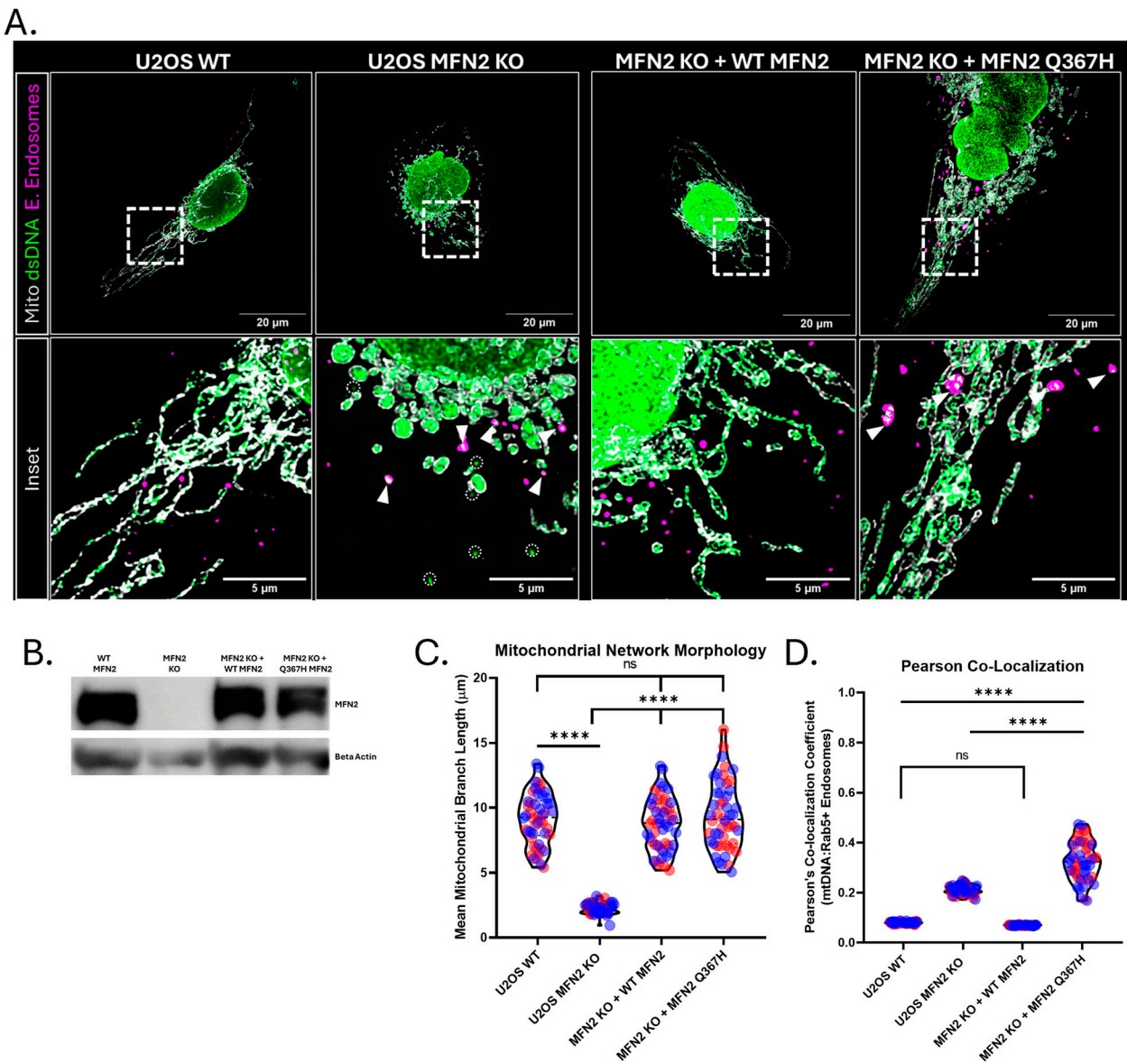

**Figure 9. Re-Expression of WT MFN2 and MFN2 Q367H in KO cells.**
**(A)** Representative confocal images showing mitochondria (TOMM20; grey), mtDNA (dsDNA; green), and early endosomes (Rab5; magenta) in U2OS WT, U2OS MFN2 KO, and U2OS MFN2 KO cells with re-expression of WT MFN2 and U2OS MFN2 KO cells with re-expression of Q367H MFN2 (left to right). Insets are of the boxed regions in above pictures, dotted circles show mtDNA outside the mitochondrial network, whereas arrows show mtDNA outside the mitochondrial network and colocalized with early endosomes. **(B)** Western blot analyses confirming the re-expression of WT MFN2 and MFN2 Q367H in U2OS MFN2 KO cells. **(C)** Quantitative analyses of mitochondrial network morphology in the four cell lines. **(D)** Quantitative analyses of Pearson's colocalization coefficient between mtDNA and early endosomes in the four cell lines. Colours indicate biological replicates; violins show the median and interquartile range. Statistics: two-way ANOVA (ns = not significant [$P > 0.05$]; *$P \leq 0.05$; **$P \leq 0.01$; ****$P \leq 0.0001$).

according to the manufacturer's protocol. Automated chip loading and templating used the Ion Chef system and 540 chip/chef kit (A30011; Thermo Fisher Scientific), and sequencing was performed on an Ion S5 system (A27212; Thermo Fisher Scientific), according to manufacturer's protocols. Base calling, read alignment to hg19, coverage analysis, and variant calling were performed with Torrent Suite (v. 5.10.1; Thermo Fisher Scientific). Similar analysis can be performed using an alternative bioinformatics pipeline as described previously (72). Patient VCFs were annotated for predicted variant consequence, gnomAD allele frequency, CADD score, and OMIM phenotypes, in addition to default parameters with Ensembl's command line Variant Effect Predictor. We in addition aligned off-target reads from exome sequencing to identify any mtDNA variants, as previously described (73).

The variant was verified by Sanger sequencing. Briefly, total genomic DNA was collected using the E.Z.N.A. Tissue DNA Extraction Kit (VWR, CA101319-018). The region of interest was amplified using PCR with the following primers: F-GAGAAGAGCAGAGAGGCTCTTG and

R-ACACAGGGAATCGTGCTGC. Subsequently, the amplified region was sequenced by Sanger sequencing at the University of Calgary Centre For Health Genomics and Informatics.

Amino acid sequence alignment was performed using the T-COFFEE online tool (74). The sequences were collected from UniProt (75). The following accession codes were used for the sequences: MFN2: O95140-1, MFN1: Q8IWA4, Mouse MFN2: Q80U63, Fish MFN2: A8WIN6, Chicken MFN2: E1BSH7, Frog MFN2: Q28CS2, Fruit fly MFN: Q7YU24, Worm MFN: Q23424.

## Quantification of protein amounts

Relative quantities were measured using Western blot analysis, as previously described (13). Briefly, cells were lysed with Pierce RIPA buffer (89900; Thermo Fisher Scientific), and subsequently, total protein was quantified by a BCA assay. 50 $\mu$g of protein was loaded from control and patient fibroblasts onto an SDS–PAGE gel. After running the gel, the blots were transferred to a PVDF membrane overnight. The blots were probed using the following antibodies: anti-MFN2 (H00009927-M03; 1:1,000; Abnova), anti-MFN1 (D6E2S; 1: 1,000; Cell Signaling Technology), anti-OPA1 (612607; 1:1,000; BD Transduction Laboratories), anti-actin (A5316; 1:1,000; Sigma-Aldrich). Secondary antibodies complimentary to the primary host species were used: goat anti-rabbit IgG HRP-linked antibody (7074S; Cell Signaling Technology) or goat anti-mouse IgG HRP (sc-2055; Santa Cruz Biotechnology). The membranes were visualized with SuperSignal West Femto Maximum Sensitivity Substrate (34095; Thermo Fisher Scientific) using Bio-Rad ChemiDoc Imaging System. Bands were quantified using Fiji (76).

## Cell culture and maintenance

Fibroblasts from the patient and healthy individual (age- and sex-matched) were generated through skin biopsies as described previously (72). The cells were cultured using Minimum Essential Media (11095080; Gibco), supplemented with 10% FBS and 1 mM sodium pyruvate. Glucose-free conditions were established using DMEM no glucose media (11966025; Gibco), supplemented with 5 mM galactose, 10% FBS, and 1 mM sodium pyruvate. Cells were grown in an incubator, maintained at 37°C and 5% $CO_2$. For the fatty acid pulse–chase assay, the cells were grown in the glucose media mentioned above, or glucose-free media with 10% FBS, but no galactose.

WT U2OS and U2OS MFN2 KO cells, a gift from Dr. Edward A Fon (7) (McGill University, Canada), were grown in DMEM high-glucose media (11965118; Gibco) supplemented with 10% FBS. A retrovirus plasmid containing the MFN2 open reading frame followed by an internal ribosomal entry site and mitochondria-targeted mNeon-Green was constructed through VectorBuilder (VB230626-1399nzw). A version of the plasmid with the Q367H variant was generated using in-fusion mutagenesis (638951; Takara Bio). Retrovirus was generated by transfecting Phoenix cells with WT or Q367H plasmids using Lipofectamine 3000 transfection (L3000015; Thermo Fisher Scientific). U2OS MFN2 KO cells were then transduced with retrovirus, and cells were sorted for mNeonGreen fluorescence using fluorescence-activated cell sorting (SH800; Sony). Endogenous levels of MFN2 expression were confirmed by immunoblotting, and these stable lines re-expressing either WT MFN2 or the Q367H variant were used for subsequent analysis.

## Mitochondrial morphology analysis

Mitochondrial morphology was analysed by immunostaining as described previously (77 Preprint), where 30,000 cells were grown in 12-mm coverslips for 24 h and subsequently fixed by 4% PFA for 20 min at 37°C. Cells were permeabilized using 0.1% Triton X-100 (J00105; MP Biomedicals) and blocked using 5% FBS. The OMM protein TOMM20 was chosen to visualize the mitochondrial network, where cells were incubated with mouse anti-TOMM20 (F-10, RRID: AB_628381; Santa Cruz Biotechnology) overnight, then subsequently probed with goat anti-mouse secondary antibody conjugated with Alexa Fluor 488 (A-11029; Thermo Fisher Scientific). Coverslips were then mounted on glass slides using ProLong Diamond Antifade Mountant with DAPI (P36966; Invitrogen).

## Oxygen consumption rate

The cellular oxygen consumption rate assay for control and patient fibroblasts was performed as previously described (78). Briefly, 40,000 cells were seeded in wells of the XF-24 plates (Agilent). The media on the plates were replaced 24 h later with Seahorse XF Base Medium, supplemented with 1 mM pyruvate, 2 mM glutamine, and 10 mM glucose, and brought to pH 7.4. Post-incubation for 45 min with $CO_2$, cells were loaded into Seahorse XF24 Analyzer. Oxygen consumption rates were measured at three time points, after administration of oligomycin (0.5 $\mu$M), carbonyl cyanide-p-trifluoromethoxyphenylhydrazone (FCCP) (1 $\mu$M), and antimycin A (0.5 $\mu$M). After the experiment, cells in each well were lysed with Pierce RIPA buffer and the total protein for each well was quantified with BCA assays, for normalization purposes.

## Measurement of mitochondrial membrane potential

Fibroblasts were stained with 400 nM TMRE (87917; Sigma-Aldrich) for 40 min, before trypsinization. Cells were then stained using LIVE/DEAD Fixable Violet Dead Cell Stain Kit (L34964; Invitrogen) and resuspended in PBS before acquisition on the cytometer. The cell fluorescence was measured using a Becton–Dickinson FACSCanto cytometer supported by BD FACSDiva software (BD Biosciences). TMRE signal was detected by a 488-nM laser with a PE detector, and live/dead signal was detected by a 405-nM laser with a BV421 detector. Dead cells and doublets were excluded from analysis. Mitochondrial membrane potential was reported as mean fluorescence intensity values of live, single cells analysed using FlowJo analysis software.

## mtDNA and mitochondrial nucleoid analyses

Mitochondrial nucleoids were visualized using confocal microscopy as previously reported (79). Briefly, 30,000 control and patient fibroblasts were grown on coverslips for 24 h and fixed with 4% PFA. Cells were permeabilized with 0.2% Triton X-100 and blocked with 5% FBS. Mitochondrial nucleoids were labelled using mouse anti-dsDNA antibody (AB_10805293, RRID:AB_10805293; Developmental

Studies Hybridoma Bank) and subsequently visualized using the goat anti-mouse secondary antibody conjugated with Alexa Fluor 488. The mitochondrial network was also labelled using anti-rabbit TOMM20 rabbit anti-TOMM20 (ab186735; Abcam), which was visualized with anti-rabbit secondary antibody conjugated with Alexa Fluor 647 (A-21245; Thermo Fisher Scientific). Coverslips were mounted using ProLong Diamond Antifade Mountant with DAPI.

mtDNA copy-number analysis was performed using qPCR. Total DNA was collected from control and patient fibroblasts seeded at the same density. Relative mtDNA copy number was analysed using nuclear-encoded housekeeping gene 18S, by amplification using the following primers:

mito-F- CACCCAAGAACAGGGTTTGT.
mito-R- TGGCCATGGGTATGTTGTTAA.
18S-F- TAGAGGGACAAGTGGCGTTC.
18S-R- CGCTGAGCCAGTCAGTGT.

For qPCR, 50 ng of total DNA was loaded into each well of a 96-well plate and the reaction was performed using the QuantStudio 6 Flex Real-Time PCR system (Thermo Fisher Scientific). The delta Ct method was used to determine the relative copy number.

Long-range PCR was done to look for mtDNA deletions using the Takara LA Taq polymerase (RR002M; Takara Bio). Total DNA was collected from control and patient fibroblasts, and the 16.2-kBP fragment was amplified using the following primers: F-ACCGCCC-GTCACCCTCCTCAAGTATACTTCAAAGG and R-ACCGCCAGGTCCTTTGAGT-TTTAAGCTGTGGCTCG. Long-range PCR was done using the following conditions: 94°C for 1 min; 98°C for 10 s and 68°C for 11 min [30 cycles], followed by a final extension cycle at 72°C for 10 min. The products after the reaction were visualized by gel electrophoresis on a 0.6% gel run for 18 h at 18V.

## Mito-ER contact sites

The proximity ligation assay was used to look for MERCs, using the Duolink In Situ Detection kit (DUI92008; Sigma-Aldrich). Briefly, 30,000 control and patient cells were grown on coverslips for 24 h and subsequently fixed using 4% PFA. Cells were permeabilized with 0.1% Triton X-100 and blocked using the blocking reagent provided by the kit. The cells were then incubated overnight using mouse anti-calnexin (MAB3126; Sigma-Aldrich) and rabbit anti-TOMM20 (ab186735; Abcam). After incubation, PLA probes, complementary to the primary antibodies, anti-mouse (DUO92004; Sigma-Aldrich) and anti-rabbit (DUO92002; Sigma-Aldrich), were added to the coverslips. The addition of DNA ligase and DNA polymerase was subsequently carried out using manufacturer's instructions and using the reagents provided in the kit. Finally, anti-mouse secondary antibody conjugated with Alexa Fluor 488 (A-11029; Thermo Fisher Scientific) and anti-rabbit secondary antibody conjugated with Alexa Fluor 647 (A-21245; Thermo Fisher Scientific) were added for visualizing the ER and mitochondrial networks, respectively. The coverslips were mounted with ProLong Diamond Antifade Mountant with DAPI before imaging.

## Lipid droplets and fatty acid pulse–chase

Total cellular lipid droplets were labelled using HCS LipidTOX Green (H34350; Thermo Fisher Scientific), following the same immunostaining

protocol to label the mitochondrial network, except for using 0.1% saponin (SAE0073-10G; Sigma-Aldrich) as the permeabilizing solution. The mitochondrial network was also labelled using anti-rabbit TOMM20 rabbit anti-TOMM20 (ab186735; Abcam), which was visualized with anti-rabbit secondary antibody conjugated with Alexa Fluor 647 (A-21245; Thermo Fisher Scientific).

For the fatty acid pulse–chase assay, 50,000 cells were seeded on a 35-mm glass-bottom dish with a 20-mm indented imaging region. Cells were grown for 24 h, after which they were pulsed with the fluorescent fatty acid precursor BODIPY 558/568 $C_{12}$ (D3835; Invitrogen). After 24 h of incubation with the BODIPY, cells were incubated with either glucose or glucose-free media and chased at time points 0, 6, and 24 h after the addition of the two medium conditions. When the variable medium conditions were added, MitoTracker Green (M7514; Invitrogen) was also added at 50 nM concentration for visualizing the mitochondrial network.

## Mitochondria, mtDNA, and early endosome imaging

The triple staining of the mitochondrial network, mtDNA, and early endosomes was performed using sequential immunostaining. The steps until blocking are the same as described above for the mitochondrial network. After blocking, primary antibodies were added for the mtDNA (same as above) and mitochondrial network (same as above) and secondary antibodies were added accordingly: anti-mouse secondary antibody conjugated with Alexa Fluor 488 (A-11029; Thermo Fisher Scientific) and anti-rabbit secondary antibody conjugated with Alexa Fluor 568 (A-11004; Thermo Fisher Scientific). After secondary antibody incubation, coverslips were thoroughly washed with PBS and the primary antibody to label early endosomes was added: rabbit anti-Rab5 (3547T; Cell Signaling Technology). Subsequently, the complementary anti-rabbit secondary antibody conjugated with Alexa Fluor 647 (A-21245; Thermo Fisher Scientific) was added. The coverslips were mounted using ProLong Diamond Antifade Mounting Media.

## Fibroblast transdifferentiation

The coding sequence for human myoD1 (accession NM_002478) was PCR-amplified from a human skeletal muscle cDNA library and subcloned into pWPXL (pWPXL was a gift from Didier Trono [plasmid # 12257; Addgene] in place of eGFP at the 5'BamHI and 3'EcoRI restriction sites). The DNA sequence was confirmed using Sanger DNA sequencing.

Lentivirus production was performed using 293FT cells grown to 90% confluency on five, 15-cm$^2$ cell culture plates, by cotransfecting 112.5 μg pWPXL-myoD1, 73 μg psPAX2, and 39.5 μg pMD2.G (gifts from Didier Trono [plasmid # 12260; Addgene] using PEIpro transfection reagent [Polyplus]). The cell culture media were collected 48 h post-transfection, quantified using Lenti-X GoStix Plus (631280; Takara Bio), centrifuged at 500$g$ for 5 min, 0.45-μm filtered, and stored at −80°C as 5 ml aliquots.

Non-immortalized primary fibroblasts below passage 5 were grown at 37°C, 5% $CO_2$ in complete media: DMEM—high glucose (10-013-CV; Corning), 10% FBS (35-077-CV; Corning), 1% P/S (VWR, K952). T75s or 24-well plates were coated in 0.1 mg/ml VitroCol Type I human collagen (5008; Advanced BioMatrix) for 1 h before gently

washing with PBS. Cells were seeded at 6,300/cm$^2$ and transduced at 75% confluency. 5 $\mu$g/ml polybrene (TR-1003-G; Sigma-Aldrich) was mixed with MYOD1 lentivirus at GoStix values of either 11,500 or 12,000 depending on the biological replicate. Cells were transduced by spinfection, specifically a 1-h spin at 800$g$, RT, and then, the lentiviral supernatant was replaced with complete media. Media were changed every 2 d, and cells were allowed to trans-differentiate for 7 d before fixing or RNA extraction. Conversion to myoblasts was confirmed by visualization of desmin using desmin (D93F5) XP rabbit mAb (cat. #5332S; CST) and Alexa Fluor 594 chicken anti-rabbit secondary antibody (A-21442; Invitrogen). RNA for qRT-PCR was extracted using RNeasy Mini Kit (74104; QIAGEN).

### Analysis of TLR9-NF-$\kappa$B signalling pathways

The expression of genes associated with the TLR9-NF-$\kappa$B was analysed by qRT-PCR. The TLR9 inhibitor chloroquine (C973P63; MilliporeSigma) and the cGAS-STING inhibitor Ru.521 (AOB37877; Aobious) were added to the cell culture media (final concentration 1 $\mu$M) for 24 h, and subsequently, the cells were washed five times with PBS. Briefly, total RNA was extracted from cells using HP Total RNA Extraction Kit (CA101414-852; VWR), according to the manufacturer's instructions. The collected RNA was reverse-transcribed using iScript Advanced cDNA Synthesis Kit (1725038; Bio-Rad). Subsequently, qPCR was performed using the QuantStudio 6 Real-Time PCR system (Thermo Fisher Scientific). The primers used for amplification of target genes are as follows:

Beta Actin-F-AAGACCTGTACGCCAACACA.
Beta Actin-R-AGTACTTGCGCTCAGGAGGA.
Asc-F-CATGAACTGATCGACAGGATG.
Asc-R-GGACCTCCTCCAAATGTTTC.
IL6-F-AGACAGCCACTCACCTCTTCAG.
IL6-R-TTCTGCCAGTGCCTCTTTGCTG.
Nlrp3-F-GCTGGCATCTGGATGAGGAA.
Nlrp3-R-GTGTGTCCTGAGCCATGGAA.
S100a9-F-GGAATTCAAAGAGCTGGTGC.
S100a9-R-TCAGCATGATGAACTCCTCG.
TNF-F-GCCCATGTTGTAGCAAACCC.
TNF-R-GGAGGTTGACCTTGGTCTGG.
IFN-$\alpha$-F-AAATACAGCCCCTTGTGCCTGG.
IFN-$\alpha$-R-GGTGAGCTGGCATACGAATCA.
IFN-$\beta$-F-AAGGCCAAGGAGTACAGTC.
IFN-$\beta$-R-ATCTTCAGTTTCGGAGGTAA.

### Imaging and analyses

All imaging was performed as previously described (24). The Olympus Spinning Disk Confocal System (Olympus SD-OSR) was used with the UAPON 100XOTIRF/1.49 oil objective lens. For live-cell imaging, a cellVivo incubation module was used to maintain cells at 37°C and 5% $CO_2$. For analysis of mitochondrial branch length and footprint, the MiNA plugin (80) was used in Fiji. The number and size of PLA puncta, mitochondrial nucleoids, lipid droplets, and endosomes were performed using the particle analysis tool on Fiji, after thresholding. All colocalization studies were performed using the JaCoP plugin (81) in Fiji. Extra-mtDNA nucleoids were quantified using the mitoQC plugin on Fiji (82), setting the mtDNA as "red"

signals and mitochondria as "green" signals. Mito-endosome distances were calculated by quantifying average linear distance between respective signals. All quantitative figures and statistical tests were performed using GraphPad Prism 9.

## Supplementary Information

## Acknowledgements

The authors would like to thank the study participants and their family. Lentivirus production was performed by the Hotchkiss Brain Institute Molecular Core Facility at the University of Calgary. FACS sorting was performed at the Flow Cytometry Core Facility at the University of Calgary. This work was supported by funds provided by the Alberta Children's Hospital Research Institute (TE Shutt), the Canadian Institutes of Health Research (TE Shutt), and the New Frontiers Research Fund (TE Shutt). M Zaman was supported by a Hotchkiss Brain Institute International Graduate Recruitment Scholarship. W Almutawa was supported by the Saudi Cultural Bureau. R Sabouny was supported by a QEII Graduate Scholarship. TGB Soule was supported by an Alberta Graduate Excellence Scholarship and a University of Calgary Faculty of Graduate Studies Doctoral Scholarship. M Joel is the recipient of the Katharine Sarah Melinda Mei-Ling Thomas Rare Diseases Scholarship. Infrastructure in the laboratory of G Pfeffer for this work was supported by the HBI Dementia Research Equipment Fund and a Canada Foundation for Innovation JELF Grant (Neuromuscular Genetics, 36624). The funders had no role in study design, data collection and interpretation, or the decision to submit the work for publication.

### Author Contributions

M Zaman: conceptualization, data curation, formal analysis, validation, investigation, visualization, methodology, and writing—original draft, review, and editing.
G Sharma: conceptualization, formal analysis, validation, investigation, visualization, and methodology.
W Almutawa: conceptualization, data curation, formal analysis, investigation, methodology, and writing—original draft.
TGB Soule: investigation and methodology.
R Sabouny: investigation and methodology.
M Joel: data curation, formal analysis, and investigation.
A Mohan: investigation and methodology.
C Chute: investigation and methodology.
JT Joseph: data curation, formal analysis, and investigation.
G Pfeffer: conceptualization, data curation, formal analysis, supervision, funding acquisition, investigation, and writing—original draft, review, and editing.
TE Shutt: conceptualization, supervision, funding acquisition, visualization, project administration, and writing—original draft, review, and editing.

### Conflict of Interest Statement

The authors declare that they have no conflict of interest.

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
