## [Reviewer comments · Life Science Alliance]

Life Science Alliance

The MFN2 Q367H variant reveals a novel pathomechanism connected to mtDNA-mediated inflammation.

Mashiat Zaman, Govinda Sharma, Walaa Almutawa, Tyler Soule, Rasha Sabouny, Matthew Joel, Armaan Mohan, Cole Chute, Jeffrey Joseph, Gerald Pfeffer, and Timothy Shutt

DOI: <https://doi.org/10.26508/lsa.202402921>

Corresponding author(s): Timothy Shutt, University of Calgary and Gerald Pfeffer, University of Calgary

Review Timeline:

Submission Date:	2024-07-02
Editorial Decision:	2024-08-05
Revision Received:	2025-02-19
Editorial Decision:	2025-02-27
Revision Received:	2025-03-14
Accepted:	2025-03-17

Transaction Report:

August 5, 2024

Re: Life Science Alliance manuscript #LSA-2024-02921-T

Dr. Timothy E Shutt
University of Calgary
Medical Genetics
HMRB 268
3330 University Dr. NW
Calgary T2N 4N1
Canada

Dear Dr. Shutt,

Thank you for submitting your manuscript entitled "The MFN2 Q367H variant reveals a novel pathomechanism connected to mtDNA-mediated inflammation" to Life Science Alliance. The manuscript was assessed by expert reviewers, whose comments are appended to this letter. We invite you to submit a revised manuscript addressing the Reviewer comments.

Thank you for this interesting contribution to Life Science Alliance. We are looking forward to receiving your revised manuscript.

Sincerely,

B. MANUSCRIPT ORGANIZATION AND FORMATTING:

Reviewer #1 (Comments to the Authors (Required)):

Zaman and colleagues report a rare case of myopathy compatible with Charcot-Marie-Tooth type 2A (CMT2A) diagnosis, as the patient carries a missense mutation in the MFN2 gene (Q367H). The authors perform a battery of cell biology experiments in patient-derived fibroblasts and demonstrate clear affections on mitochondrial dynamics and metabolism mainly when exposed to conditions leading to energy stress (galactose medium), and connect these observations to a potential dysfunction of the MFN2Q367H protein. The authors observe the release of mtDNA in the cytoplasm and in early endosomes, which is associated with the activation of inflammation signaling pathways. The work shows some mechanistic gaps when demonstrating the link between mitochondrial aberrations and MFN2Q367H. Additionally, the role of mtDNA-mediated inflammation in driving the observed cellular phenotype is not fully addressed.

Major concerns:

1. I agree that the observed mitochondrial features could easily be attributed to the malfunctioning of a MFN2 variant. However, patient fibroblasts come with an unknown genetic background that could contribute to the phenotype beyond MFN2, limiting the authors' capability to claim that MFN2Q367H is a "standalone disease causing variant" (line 439). To obtain prove-of-concept evidence on MFN2Q367H pathogenicity, the authors should perform one of the following experiments or similar: 1) overexpress the MFN2Q367H variant in healthy donor fibroblast and see if it recapitulates some of the mitochondrial features observed in patient ones; 2) perform a rescue experiment by overexpressing a wild-type MFN2 protein in patient fibroblast (as done by other authors for MFN1; PMID: 30882371) and check if mitochondrial alterations are corrected. In my opinion, this type of experiment would increase the quality of the work by demonstrating that MFN2Q367H alone is sufficient to promote mitochondrial alterations in cells.
2. OCR data (Figure 2D) show that patient-derived fibroblasts have a higher leak but lower maximal respiratory capacity. Higher leak implies that mitochondria of MFN2Q367H cells are highly uncoupled. I would recommend assessing this possibility using a potential-sensitive dye such as MitoTracker Red CMXRos or TMRM and quantifying changes in the signal by flow cytometry (PMID: 21486251). Additionally, reduced maximal respiration indicates decreased capacity/inefficiency of the electron transport chain, which could lead to increased oxidative stress.
3. Patient fibroblasts show a normal mitochondrial network in the presence of glucose but fragment upon galactose treatment (Figure 2A). This change in shape does not occur in control cells. Are patient-derived cells able to cope with galactose as the main energy source or do they activate programs of apoptosis?
4. The number of MERCs detected by PLA is remarkably high and spread throughout the cell, including zones where neither mitochondria nor ER staining is present (Figure 4A). The authors do not provide any technical controls to discard unspecific binding of the primary antibodies. Was the protocol validated by using cells KD for either TOMM20 or Calnexin (either by the authors or in published literature)? Each individual PLA point equals to a single TOMM20-Calnexin proximity. How can MERCs size be inferred from this type of data (Figure 4C)?
5. Patient fibroblasts show activation of mtDNA-mediated inflammation, which can be attenuated by using chloroquine and RU.521 (Figure 6A). In this regard, are mitochondrial alterations (e.g. morphology) corrected when mtDNA inflammation is blocked using these inhibitors?

Minor comments and suggestions:

1. The authors refer to a muscle biopsy taken from the patient and describe a "necrotic and regeneration myofiber, myofiber hypertrophy, multifocal endomysia, and perimysial fibrosis, and extensive infiltration by adipose tissue. Immunostaining for dystrophin, alpha/gamma sarcoglycan, sceptrin, α laminin, desmin, and α β 294 crystallin was unremarkable. There was no

inflammatory cell infiltrate" (page 11, line 292). Representative images displaying all these characteristics should be added to the manuscript.

2. The authors use MRI to examine muscle size in the upper and lower regions of the legs and claim that the patient suffers from muscle atrophy. How did the authors reach this diagnosis? Which are reference values that distinguish a healthy patient from one with muscle atrophy?

3. Were control fibroblasts originating from a healthy donor with a similar age, ethnic background, and similar comorbidities as the patient? Please indicate any information you might have on the healthy individual.

4. Given the decrease in MERCs in patient fibroblasts, I would suggest assessing some markers of unfolded protein response (URP) activation such as increased transcription of ATF4 target genes.

Reviewer #2 (Comments to the Authors (Required)):

The study investigates a novel MFN2 variant, Q367H, identified in a patient with late-onset distal myopathy but lacking peripheral neuropathy. This variant results in a glutamine to histidine substitution, suggesting a previously uncharacterized impact on MFN2's functions. Key findings include altered mitochondrial DNA nucleoid distribution and reduced mitochondria-endoplasmic reticulum contacts in patient-derived cells, indicating impaired MFN2 functions. Notably, the MFN2 Q367H variant is linked to increased activation of TLR9 and cGAS-STING inflammation pathways, suggesting a novel pathomechanism involving mtDNA-mediated inflammation that contributes to myopathy. These findings support the pathogenic potential of the Q367H variant, highlighting the complex role of MFN2 in cellular processes beyond mitochondrial fusion and emphasizing the need for further research into therapeutic strategies for MFN2-related disorders. The paper is well-written and contributes valuable new data to the understanding of MFN2-associated diseases and the role of mtDNA-mediated inflammation in neuromuscular disorders. By identifying the MFN2 Q367H variant and elucidating its impact on mitochondrial dynamics and inflammation pathways, the authors provide insights into the complex mechanisms underlying myopathies and neuropathies linked to MFN2 dysfunction. This research enhances the current knowledge of how mitochondrial and endoplasmic reticulum interactions are altered in pathological states, furthering the scientific community's understanding of these processes in the context of muscle disease. However, additional proofreading is warranted as there are typos and minor slips throughout the manuscript (e.g., line 548). A question remains whether electron microscopy was performed on the patient's skeletal muscle to assess mitochondrial ultrastructure, as this could provide further insight into the morphological impacts of the MFN2 Q367H variant.

Response to Reviews

We appreciate the constructive feedback and comments from the reviewers, which we have taken to heart and addressed in our revised manuscript, as detailed in the point-by-point responses below in red. Specifically, we have added new data showing that when expressed in a different cell line, the MFN2 Q367H variant is sufficient to induce mtDNA release. This new finding addresses the primary concern of reviewer #1 regarding the potential complications from genetic background. In addition, as requested, we have provided new data showing that there is no change in membrane potential in patient fibroblasts, and analysis validating the use of the PLA assay to measure MERCs. Finally, we have also added new clinical data with respect to the patient's muscle biopsy. Together, we feel that these new data further strengthen the argument that the MFN2 Q367H variant is causing the patient phenotypes via mtDNA-mediated inflammation. As such, we would like to thank the reviewers for their valuable feedback, which helped to improve the manuscript.

Reviewer #1 (Comments to the Authors (Required)):

Zaman and colleagues report a rare case of myopathy compatible with Charcot-Marie-Tooth type 2A (CMT2A) diagnosis, as the patient carries a missense mutation in the MFN2 gene (Q367H). The authors perform a battery of cell biology experiments in patient-derived fibroblasts and demonstrate clear affections on mitochondrial dynamics and metabolism mainly when exposed to conditions leading to energy stress (galactose medium), and connect these observations to a potential dysfunction of the MFN2Q367H protein. The authors observe the release of mtDNA in the cytoplasm and in early endosomes, which is associated with the activation of inflammation signaling pathways. The work shows some mechanistic gaps when demonstrating the link between mitochondrial aberrations and MFN2Q367H. Additionally, the role of mtDNA-mediated inflammation in driving the observed cellular phenotype is not fully addressed.

Major concerns:

1. I agree that the observed mitochondrial features could easily be attributed to the malfunctioning of a MFN2 variant. However, patient fibroblasts come with an unknown genetic background that could contribute to the phenotype beyond MFN2, limiting the authors' capability to claim that MFN2Q367H is a "standalone disease causing variant" (line 439). To obtain prove-of-concept evidence on MFN2Q367H pathogenicity, the authors should perform one of the following experiments or similar: 1) overexpress the MFN2Q367H variant in healthy donor fibroblast and see if it recapitulates some of the mitochondrial features observed in patient ones; 2) perform a rescue experiment by

overexpressing a wild-type MFN2 protein in patient fibroblast (as done by other authors for MFN1; PMID: 30882371) and check if mitochondrial alterations are corrected. In my opinion, this type of experiment would increase the quality of the work by demonstrating that MFN2Q367H alone is sufficient to promote mitochondrial alterations in cells.

Thank you for the suggestion. We agree that using complementary system to look at the pathogenicity of MFN2 Q367H would provide additional confidence that this variant is causing the functional alterations we report in patient fibroblasts. To this end, we set up a system that allows us to re-express MFN2 (WT or Q367H) in an MFN2 KO line. Notably, cells re-expressing MFN2 Q367H, but not WT MFN2, phenocopy our observations in patient fibroblasts (New Figure 9). In particular, we observed the same DNA release phenotype in the Q367H expressing cells. These new data provide direct evidence that the MFN2 Q367H variant is sufficient to cause the mtDNA release.

2. OCR data (Figure 2D) show that patient-derived fibroblasts have a higher leak but lower maximal respiratory capacity. Higher leak implies that mitochondria of MFN2Q367H cells are highly uncoupled. I would recommend assessing this possibility using a potential-sensitive dye such as MitoTracker Red CMXRos or TMRM and quantifying changes in the signal by flow cytometry (PMID: 21486251). Additionally, reduced maximal respiration indicates decreased capacity/inefficiency of the electron transport chain, which could lead to increased oxidative stress.

As requested, we have measured membrane potential, but do not see any statistically significant changes between control and patient fibroblasts. Ultimately, while the differences in OCR are intriguing, this is not a major focus of our study and we are not making any claims on these differences or their underlying mechanisms, we are simply reporting the differences.

3. Patient fibroblasts show a normal mitochondrial network in the presence of glucose but fragment upon galactose treatment (Figure 2A). This change in shape does not occur in control cells. Are patient-derived cells able to cope with galactose as the main energy source or do they activate programs of apoptosis?

This is an important point raised by the reviewer. In our observations, we do not see any obvious differences in growth rates or confluence of the patient fibroblasts within the timeframe that the experiments are being conducted, suggesting that the cells are indeed able to cope in galactose media, even if their mitochondria are sub-par. Ultimately, the main phenomenon that we are focused on in the paper (mtDNA release),

occurs regardless of the growth conditions. As such, exploring any potential differences between glucose and galactose media is beyond the scope of the current research.

4. The number of MERCs detected by PLA is remarkably high and spread throughout the cell, including zones where neither mitochondria nor ER staining is present (Figure 4A). The authors do not provide any technical controls to discard unspecific binding of the primary antibodies. Was the protocol validated by using cells KD for either TOMM20 or Calnexin (either by the authors or in published literature)? Each individual PLA point equals to a single TOMM20-Calnexin proximity. How can MERCs size be inferred from this type of data (Figure 4C)?

The in-situ proximity ligation assay is a common technique to interrogate mito-ER contact sites [PMID: 28060261 / 34926468 / 37292700]. As we mention in the main text, this assay is only a proxy for studying MERCs, not a direct analysis of every contact site. To address concerns about the PLA assay, it is important to note that to reduce background signal we use low concentrations of the hybridizing probes to avoid off target effects, as recommended [PMID: 29339495]. Moreover, we have added new analysis validating the PLA assay (Sup Fig 1), which measures colocalization between the PLA signal and either the mitochondrial signal or the ER signal (similar to PLA analysis in PMID: 35620072). In both cases, there is a strong co-localization (>0.9 Pearson's Coefficient) in all samples. This new analysis shows that the vast majority of the PLA signals are in fact where we would expect to find them in the case of mito-ER contact sites. While there is a small amount of non-specific signal, this amount is the same across all samples and does not impact the findings when we compare the signal between samples. If anything, background signal would make these differences less pronounced.

5. Patient fibroblasts show activation of mtDNA-mediated inflammation, which can be attenuated by using chloroquine and RU.521 (Figure 6A). In this regard, are mitochondrial alterations (e.g. morphology) corrected when mtDNA inflammation is blocked using these inhibitors?

We agree with the reviewer that whether mtDNA-mediated inflammation impacts mitochondrial morphology is an intriguing question. However, the fact that we do not see morphology changes in patient cells (both fibroblasts in glucose or myoblasts), when inflammation is present, argues against inflammation driving changes in morphology. Moreover, any interpretation of whether attenuating inflammation with these drugs impacts mitochondrial alterations would be confounded by the potential for off-target effects of the drugs and the multiple functions performed by MFN2. Thus, more complex

experiments that are beyond the scope of this study would be required. As such, in this paper we are not trying to link the inflammation to the mitochondrial morphology/alterations.

Minor comments and suggestions:

1. The authors refer to a muscle biopsy taken from the patient and describe a "necrotic and regeneration myofiber, myofiber hypertrophy, multifocal endomyasia, and perimysial fibrosis, and extensive infiltration by adipose tissue. Immunostaining for dystrophin, alpha/gamma sarcoglycan, sceptrin, α laminin, desmin, and α β 294 crystallin was unremarkable. There was no inflammatory cell infiltrate" (page 11, line 292). Representative images displaying all these characteristics should be added to the manuscript.

In response to the reviewer's request, we have included new data of the pathologic changes on muscle biopsy (Figure 1A). Accompanying this figure, we have included a detailed figure legend explaining the findings: "Representative images of the muscle biopsy (A-F) including two formalin-fixed, paraffin-embedded slides (panels A and F) and four snap-frozen muscle sections (panels B-E). Distances are represented in the scale bars. Panel A (hematoxylin-eosin; H&E) demonstrates extensive adipose tissue ("fatty replacement") in the biopsy (black arrow). This was less obvious in the frozen tissue, which had been dissected away from obvious fat. Panel B (H&E) illustrates the variation in myofiber size, as well as the increased connective tissue between myofibers (endomysial fibrosis; green arrow). The biopsy had several necrotic fibers (panel C H&E; black arrow) and increased numbers of internalized nuclei (black arrowhead). In panel D (H&E) are two basophilic regenerating fibers (black arrows), internalized nuclei (black arrowheads), and increased endomysial connective tissue (green arrow). Dystrophin immunoperoxidase stains the myofiber sarcolemma (Dys2 illustrated in panel E). Dys1 and Dys3 are similar (data not shown). The desmin immunoperoxidase in longitudinal sections (panel F) illustrates the repetitive sarcolemmal units (stripes in each fiber) but does not stain significant sarcoplasmic deposits."

2. The authors use MRI to examine muscle size in the upper and lower regions of the legs and claim that the patient suffers from muscle atrophy. How did the authors reach this diagnosis? Which are reference values that distinguish a healthy patient from one with muscle atrophy?

Thank you for this important comment. The initial description provided in the text of “muscle atrophy” was not sufficiently precise and we have modified this to correctly state “fatty infiltration” of the same muscles. The finding is based on the T1 MRI findings, for which we have shown representative images. The high signal abnormality demonstrated in adductor longus, gluteus minimus, tibialis anterior, and inferior gastro-soleus bilaterally is presumed due to chronic fibroadipose replacement of muscle tissues. There was no corresponding signal abnormality in T2 imaging, which suggests that active inflammation (myositis) was not present.

3. Were control fibroblasts originating from a healthy donor with a similar age, ethnic background, and similar comorbidities as the patient? Please indicate any information you might have on the healthy individual.

As requested, we have added information on the origin of the healthy control fibroblasts, which are from an age and sex matched individual.

4. Given the decrease in MERCs in patient fibroblasts, I would suggest assessing some markers of unfolded protein response (URP) activation such as increased transcription of ATF4 target genes.

Although the question raised by the reviewer is interesting, it is beyond the scope of our current work. Notably, we are actively researching links between the integrated stress response and MFN2 dysfunction [PMID: 38915623] and plan to look at this link with respect to the MFN2 Q367H variant in the future.

Reviewer #2 (Comments to the Authors (Required)):

The study investigates a novel MFN2 variant, Q367H, identified in a patient with late-onset distal myopathy but lacking peripheral neuropathy. This variant results in a glutamine to histidine substitution, suggesting a previously uncharacterized impact on MFN2's functions. Key findings include altered mitochondrial DNA nucleoid distribution and reduced mitochondria-endoplasmic reticulum contacts in patient-derived cells, indicating impaired MFN2 functions. Notably, the MFN2 Q367H variant is linked to increased activation of TLR9 and cGAS-STING inflammation pathways, suggesting a novel pathomechanism involving mtDNA-mediated inflammation that contributes to myopathy. These findings support the pathogenic potential of the Q367H variant, highlighting the complex role of MFN2 in cellular processes beyond mitochondrial fusion and emphasizing the need for further research into therapeutic strategies for MFN2-related disorders. The paper is well-written and contributes valuable new data to the

understanding of MFN2-associated diseases and the role of mtDNA-mediated inflammation in neuromuscular disorders. By identifying the MFN2 Q367H variant and elucidating its impact on mitochondrial dynamics and inflammation pathways, the authors provide insights into the complex mechanisms underlying myopathies and neuropathies linked to MFN2 dysfunction. This research enhances the current knowledge of how mitochondrial and endoplasmic reticulum interactions are altered in pathological states, furthering the scientific community's understanding of these processes in the context of muscle disease.

However, additional proofreading is warranted as there are typos and minor slips throughout the manuscript (e.g., line 548).

Thank you for pointing this out. We will endeavour to fix these minor issues before final publication.

A question remains whether electron microscopy was performed on the patient's skeletal muscle to assess mitochondrial ultrastructure, as this could provide further insight into the morphological impacts of the MFN2 Q367H variant.

We agree that including electron microscopy would have provided valuable additional information. Unfortunately, the biopsy was not processed for electron microscopy. We have acknowledged this limitation in the text, to emphasize the added value of electron microscopy in the investigation of further cases like these: "Future cases of myopathy associated with this variant in *MFN2* may benefit from the inclusion of electron microscopic examination of muscle biopsy tissues. EM was not obtained for our case, but may have shown mitochondrial ultrastructural changes to provide further insight into the morphological impacts of the of the *MFN2* variant."

February 27, 2025

RE: Life Science Alliance Manuscript #LSA-2024-02921-TR

Dr. Timothy E Shutt
University of Calgary
Medical Genetics
HMRB 268
3330 University Dr. NW
Calgary T2N 4N1
Canada

Dear Dr. Shutt,

Thank you for submitting your revised manuscript entitled "The MFN2 Q367H variant reveals a novel pathomechanism connected to mtDNA-mediated inflammation.". We would be happy to publish your paper in Life Science Alliance pending final revisions necessary to meet our formatting guidelines.

- please address the Reviewer's remaining comment #2
- please be sure that the authorship listing and order is correct
- please upload your Tables in editable .doc or Excel format
- please consult our manuscript preparation guidelines <https://www.life-science-alliance.org/manuscript-prep> and make sure your manuscript sections are in the correct order
- please add authors' contributions to our system as well
- please add your main, supplementary figure, and table legends to the main manuscript text after the references section
- you may remove panel A from figures S1 and S2 and their legends, as they have only one panel
- please add callouts for Figures 1B-F, 8C, and 9B, D to your main manuscript text

LSA now encourages authors to provide a 30-60 second video where the study is briefly explained. We will use these videos on social media to promote the published paper and the presenting author (for examples, see <https://docs.google.com/document/d/1-UWCfbE4pGcDdcgzcmiuJl2XMBJnxKYeqRvLLrLS08s/edit?usp=sharing>). Corresponding or first-authors are welcome to submit the video. Please submit only one video per manuscript. The video can be emailed to contact@life-science-alliance.org

A. FINAL FILES:

B. MANUSCRIPT ORGANIZATION AND FORMATTING:

Thank you for your attention to these final processing requirements. Please revise and format the manuscript and upload materials within 5 days.

Sincerely,

Reviewer #1 (Comments to the Authors (Required)):

1. The authors have provided new data that supports their initial claims. In particular, the experiment where MFN2 WT and Q367H were repressed in MFN2-KO cells confirms that MFN2 mutation variant recapitulates the phenotype observed in the patient's fibroblast.
2. The authors have answered my concerns on the high amount of PLA points with literature. However, the references provided do not involve the combination of primary antibodies used by the authors (TOMM20+Calnexin), which was my main concern when referring to unspecific binding. Beyond that, I still do not understand how MERC size can be inferred from a test that measures proximity between two proteins, each discrete point being an individual interaction. I do not recall any publication using PLA data to measure this parameter either. Can you please elaborate on what you understand as "size"?
3. Finally, I want to point out that high concentrations of TMRM (such as 400 nM) lead to signal quenching and are not recommended for determining differences among two static populations (PMID: 21486251). Therefore, the experiment should be repeated using lower concentrations (10-30 nM). However, since the authors stress that "this is not a major focus of the study", I leave it to the editor to decide whether to repeat the assay. I disagree with the authors on this point since membrane potential is a highly relevant feature of mitochondrial bioenergetics, as would be assessing if patient fibroblasts have increased levels of mitochondrial ROS.

Beyond the points above, I have no further comments or concerns.

Reviewer Comment #2:

The authors have answered my concerns on the high amount of PLA points with literature. However, the references provided do not involve the combination of primary antibodies used by the authors (TOMM20+Calnexin), which was my main concern when referring to unspecific binding. Beyond that, I still do not understand how MERC size can be inferred from a test that measures proximity between two proteins, each discrete point being an individual interaction. I do not recall any publication using PLA data to measure this parameter either. Can you please elaborate on what you understand as "size"?

Response: We thank the reviewer for their positive feedback on the revised manuscript. Here we provide additional context and clarification regarding the PLA assay. The references we provided were meant to show steps used by other groups to validate PLA signal, which we have also taken for our TOMM20/Calnexin approach to estimate MERCs. To specifically validate the PLA signals, data in Sup Fig 1 show that over ~95% PLA puncta overlap with TOMM20 and Calnexin signals, which is what we would expect if the PLA signal was representative of MERCs, and thus validates the PLA signal reflects MERCs. With respect to the reviewer's question about how MERC size can be inferred from the PLA signal, we offer the following explanation, which is supported by the diagram below. A critical point of clarification here is that a single PLA puncta does not necessarily represent an individual interaction. The reason being that the resolution of confocal microscopy is ~250nm, which is insufficient to distinguish between several nearby PLA interactions. In this respect, as TOMM20 and Calnexin are abundant proteins in the outer mitochondrial membrane and the ER, respectively, there will be multiple opportunities for PLA interactions within any particular MERC. As such, the size of the puncta should be proportional to the length of the MERC (representing multiple PLA interactions) and hence can be used as a proxy for comparing the size of MERCs. As shown in the diagram, a smaller MERC will result in a smaller PLA signal, while the opposite for larger MERCs.

March 17, 2025

RE: Life Science Alliance Manuscript #LSA-2024-02921-TRR

Dr. Timothy E Shutt
University of Calgary
Medical Genetics
HMRB 268
3330 University Dr. NW
Calgary T2N 4N1
Canada

Dear Dr. Shutt,

Thank you for submitting your Research Article entitled "The MFN2 Q367H variant reveals a novel pathomechanism connected to mtDNA-mediated inflammation.". It is a pleasure to let you know that your manuscript is now accepted for publication in Life Science Alliance. Congratulations on this interesting work.

DISTRIBUTION OF MATERIALS:

Again, congratulations on a very nice paper. I hope you found the review process to be constructive and are pleased with how the manuscript was handled editorially. We look forward to future exciting submissions from your lab.

Sincerely,
